# DENSE SEMANTIC MATCHING WITH VGGT PRIOR

## ABSTRACT

Semantic matching aims to establish pixel-level correspondences between instances of the same category and represents a fundamental task in computer vision. Existing approaches suffer from two limitations: (i) Geometric Ambiguity: Their reliance on 2D foundation model features (*e.g.*, Stable Diffusion, DINO) often fails to disambiguate symmetric structures, requiring extra fine-tuning yet lacking generalization; (ii) Nearest-Neighbor Rule: Their pixel-wise matching ignores cross-image invisibility and neglects manifold preservation. These challenges call for geometry-aware pixel descriptors and holistic dense correspondence mechanisms. Inspired by recent advances in 3D geometric foundation models, we turn to VGGT, which provides geometry-grounded features and holistic dense matching capabilities well aligned with these needs. However, directly transferring VGGT is challenging, as it was originally designed for geometry matching within cross views of a single instance, misaligned with cross-instance semantic matching, and further hindered by the scarcity of dense semantic annotations. To address this, we propose an approach that (i) retains VGGT's intrinsic strengths by reusing early feature stages, fine-tuning later ones, and adding a semantic head for bidirectional correspondences; and (ii) adapts VGGT to the semantic matching scenario under data scarcity through cycle-consistent training strategy, synthetic data augmentation, and progressive training recipe with aliasing artifact mitigation. Extensive experiments demonstrate that our approach achieves superior geometry awareness, matching reliability, and manifold preservation, outperforming previous baselines.

## 1 INTRODUCTION

Semantic matching aims to establish pixel-level correspondences between semantically equivalent regions across two images with the same-category instances, which requires both low-level pixel perception and high-level semantic understanding, as shown in Fig. 1. It serves as a fundamental technique in 2D manipulation (*e.g.*, style (Cai et al., 2023) and motion (Chen et al., 2023) transfer), 3D analysis (*e.g.*, morphing (Yang et al., 2025)), and robotics (*e.g.*, affordance (Lai et al., 2021)).

Recent works (Zhang et al., 2023a; Dünkel et al., 2025) leverage off-the-shelf features from 2D foundation models, such as Stable Diffusion (Rombach et al., 2022; Tang et al., 2023) and DINO (Oquab et al., 2023), as pixel descriptors, and establish matches through nearest-neighbor search (*i.e.*, each pixel in one image is assigned to the pixel in the other image with the most similar feature). While this zero-shot paradigm shows promising results, it suffers from two limitations: (i) Geometric Ambiguity: Semantic matching is inherently a 3D problem, yet features extracted from 2D foundation models often lack explicit 3D geometry awareness, making it difficult to distinguish symmetric or repetitive patterns such as left and right eyes. Although some approaches, motivated by the desire to inject 3D priors, incorporate orientation-aligned preprocessing (Zhang et al., 2024), class-specific tuning (Mariotti et al., 2024; Barel et al., 2024; Mariotti et al., 2025), and geometric augmentation (Fundel et al., 2025) to mitigate this, such remedies remain ad hoc and struggle to generalize across diverse scenarios. (ii) Nearest-Neighbor Rule: The reliance on simple nearest-neighbor matching fails to account for cross-image invisibility (*i.e.*, source pixels lack valid counterparts in the target image) and disregards the preservation of underlying manifold structures (*i.e.*, maintaining consistent relative positions between pixels when mapped from source to target).

These long-standing limitations point to the need for models with geometry-aware pixel descriptors and holistic dense correspondence mechanisms. We observe that these desired properties naturally

Figure 1: **The motivation of leveraging VGGT priors for semantic matching and our improvements over previous approaches.** (i) Motivation: By visualizing the predicted 3D point maps, we find VGGT, designed for geometry matching between views for one instance, can coarsely align same-category instances (more cases can be found in Fig. 8). VGGT also has inherent manifold-preserving mapping properties. (ii) Improvements: Built on VGGT, our approach not only outperforms previous ones (DIY-SC (Dünkel et al., 2025), SPH (Mariotti et al., 2024), SpaceJAM (Barel et al., 2024)) in correspondence prediction but also outputs prediction confidence.

align with recent advances in 3D geometric foundation models such as DUSt3R (Wang et al., 2024) and VGGT (Wang et al., 2025), which provide strong geometry-grounded priors. Although originally developed for 3D reconstruction from images, these models inherently learn image matching capabilities as a subtask of reconstruction (Leroy et al., 2024). Our preliminary evaluation of VGGT, a state-of-the-art reconstruction model, shows that it can coarsely align instances of the same category (Fig. 1). Such properties make VGGT a highly promising basis for dense semantic matching.

However, direct transfer is challenging: VGGT was originally designed for geometry matching across views of the *same instance*, which does not fully align with the goal of semantic matching across *different instances* exhibiting variations in appearance and shape. In addition, semantic matching suffers from a lack of dense ground-truth annotations, further complicating its adaptation.

To address these challenges, we design an approach that combines capability retention and task-specific adaptation. For capability retention, we reuse VGGT's early feature stages, fine-tune its later ones, and append a semantic matching head that predicts bidirectional correspondence maps between source and target. For adaptation under limited annotations, we (i) introduce a cycle-consistent training strategy that couples matching–reconstruction consistency with error–confidence correlation; (ii) curate a synthetic data pipeline generating diverse correspondences across categories, viewpoints, and occlusions; and (iii) adopt a progressive training recipe with aliasing artifact mitigation that gradually transfers dense correspondence ability from synthetic domains to real-world data. Extensive experiments and ablation studies demonstrate the superiority of our performance and the effectiveness of each key design.

Our contributions can be summarized as follows: (i) We are the first to adapt VGGT for dense semantic matching, leveraging its priors (*i.e.*, geometry-grounded features and holistic matching capabilities) to resolve geometric ambiguities and maintain manifold-preserving mappings. (ii) We propose a cycle-consistent training strategy with matching-reconstruction consistency and error-confidence correlation, eliminating reliance on large-scale dense-annotated real data while addressing cross-image invisibility. (iii) We curate a scalable synthetic pipeline and introduce a progressive training recipe with aliasing artifact mitigation that bridges synthetic and real domains. (iv) Extensive experiments demonstrate that our approach achieves superior geometry-aware, manifold-preserving, and robust dense semantic matching compared to previous ones.

## 2 RELATED WORKS

**Learning-Based Semantic Matching.** Semantic matching is difficult due to appearance variations and scarce, ambiguous annotations (Truong et al., 2021; Zhang et al., 2025a). Early methods relied on handcrafted feature descriptors (Liu et al., 2016), while the advent of deep learning facilitated the development of more effective feature extractors (Yi et al., 2016; Kim et al., 2017; Novotny et al., 2017; Rocco et al., 2018) and direct semantic correspondence detection networks (Rocco et al., 2017; Han et al., 2017; Kim et al., 2019). To address the limitations of scarce annotations, existing

approaches have evolved three strategies: (i) leveraging weak supervision signals (Lan et al., 2021; Chen et al., 2021; Zhang et al., 2023b; Truong et al., 2022; Yang et al., 2024), (ii) employing warping and cycle consistency constraints (Zhou et al., 2016; Truong et al., 2021; 2022; Lan et al., 2022), and (iii) augmenting pseudo-label supervision (Kim et al., 2022; Li et al., 2021; Huang et al., 2023).

**Foundation-Model-Based Semantic Matching.** Recent research has demonstrated the potential of leveraging features from 2D foundation models for zero-shot semantic correspondence prediction (Amir et al., 2022; Zhang et al., 2023a; Hedlin et al., 2023; Tang et al., 2023; Cheng et al., 2024; Stracke et al., 2025). Among these, DINO features (Caron et al., 2021; Oquab et al., 2023) have been particularly effective (Amir et al., 2022; Zhang et al., 2023a; 2024; Suri et al., 2024; Fundel et al., 2025). Diffusion model features (Rombach et al., 2022; Stracke et al., 2024) offer complementary strengths (Hedlin et al., 2023; Tang et al., 2023; Zhang et al., 2023a; 2024; Mariotti et al., 2024; Li et al., 2024; Fundel et al., 2025; Xue et al., 2025). However, simple nearest-neighbor search exhibits systematic limitations in disambiguating symmetric object parts (Luo et al., 2023; Zhang et al., 2024; Mariotti et al., 2024; Li et al., 2024; Wimmer et al., 2024; Sommer et al., 2025). To address this, several studies have proposed appending adapter modules fine-tuned with pseudo or ground-truth labels (Zhang et al., 2024; Xue et al., 2025; Dünkel et al., 2025). Alternative approaches construct joint atlases for objects across multiple images (Gupta et al., 2023; Ofri-Amar et al., 2023). (Zhang et al., 2024) through keypoint-specific information, but this cannot resolve all symmetries via simple image transformations (*e.g.*, flipping) and requires keypoint-specific information that is generally unavailable. Approaches using 3D priors, such as DistillDIFT (Fundel et al., 2025), SPH (Mariotti et al., 2024), and Jamais Vu (Mariotti et al., 2025), show promise but remain limited for cross-instance and complex scenarios.

**3D Geometric Foundation Models.** Recent works (Zhang et al., 2025b) such as DUSt3R (Wang et al., 2024) and VGGT (Wang et al., 2025) replace traditional multi-stage geometry pipelines with feed-forward transformers that directly predict 3D reconstruction signals (*e.g.*, camera poses and 3D point maps). Building on these efforts, MASt3R (Leroy et al., 2024) demonstrates strong image matching performance within the same scene. In contrast, we extend this line of research from purely geometric to semantic matching, aiming to establish dense correspondences across different object instances within the same category.

## 3 METHODOLOGY

We introduce our approach from: (i) Architecture: We review VGGT and present our extension (Sec. 3.1). (ii) Training: To equip the model with dense semantic matching capability, we propose a cycle-consistent training strategy (Sec. 3.2), curate a synthetic data pipeline (Sec. 3.3), and adopt a progressive training recipe (Sec. 3.4) with aliasing artifact mitigation (Sec. 3.5).

### 3.1 SEMANTIC CORRESPONDENCE PREDICTION WITH VGGT PRIOR

**VGGT Preliminary.** Given a set of $N$ RGB images $\{I_i\}_{i=1}^N \in \mathbb{R}^{H \times W}$, VGGT applies a single feed-forward transformer backbone with $L$ blocks and DPT-based decoders (Ranftl et al., 2021), to predict camera parameters $\hat{g}_i \in \mathbb{R}^9$ and pixel-level functional maps, such as depth $\hat{D}_i \in \mathbb{R}^{H \times W}$ and 3D points $\hat{P}_i \in \mathbb{R}^{3 \times H \times W}$:

$$f : \{I_i\}_{i=1}^N \longmapsto \{(\hat{g}_i, \hat{D}_i, \hat{P}_i)\}_{i=1}^N. \tag{1}$$

Concretely, as shown in Fig. 2, each image is first patchified by a DINO encoder into tokens, which are processed by alternating frame-wise inter attention (within each image) and global cross attention (across all images) in the transformer backbone. Tokens are then reshaped into low-resolution feature maps $\hat{F} \in \mathbb{R}^{C \times H' \times W'}$ and upsampled with DPT decoders to produce $\hat{D}_i$ and $\hat{P}_i$.

**Semantic Correspondence Prediction.** Building on this pipeline, we aim to predict dense semantic correspondences between a source image $I_s$ and a target image $I_t$. As shown in Fig. 2, for the transformer backbone, we reuse the early $L_{shared}$ blocks (kept frozen) to extract geometry-grounded tokens, while fine-tuning the latter $(L - L_{shared})$ blocks to obtain semantic tokens. These are reshaped into feature maps $\hat{F}_s$, $\hat{F}_t \in \mathbb{R}^{C \times H' \times W'}$, which are fed into a new DPT-based semantic matching head $\phi_{\text{match}}$ to predict bidirectional sampling grids (*i.e.*, grids denote the coordinates used to sample color values from one image when generating the warping image):

$$\hat{G}_{s \to t} \in [-1, 1]^{2 \times H \times W}, \qquad \hat{G}_{t \to s} \in [-1, 1]^{2 \times H \times W}, \tag{2}$$

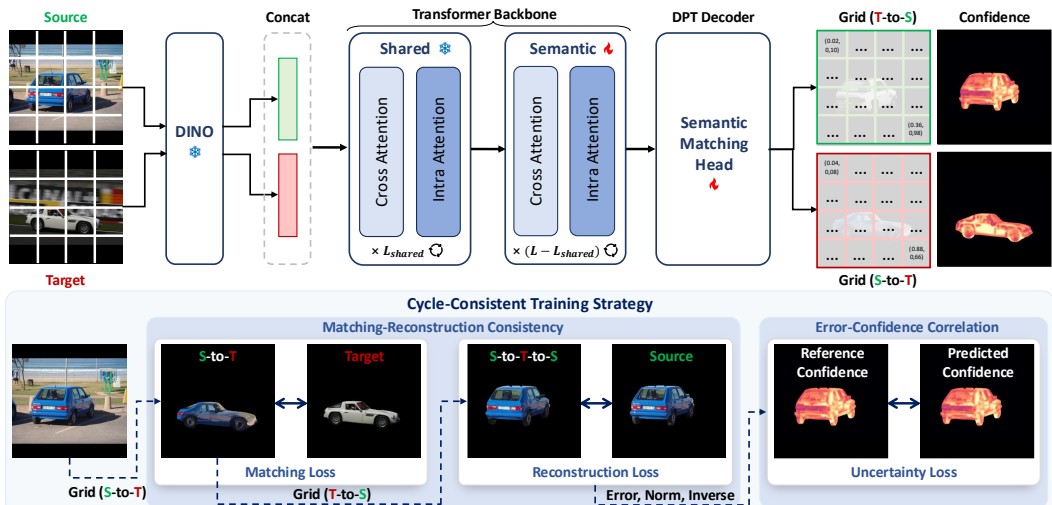

Figure 2: **Approach overview.** (i) Pipeline: The source and target images are first processed through a DINO-based feature extractor, followed by a VGGT-adapted backbone for feature refinement, and finally output pixel-wise correspondences in grid form via a semantic matching head. (ii) Cycle-Consistent Training Strategy: Matching–reconstruction consistency, ensuring that mapping an image to its counterpart and back reconstructs the original, and error–confidence correlation, aligning reconstruction error with predicted confidence to encourage reliable predictions.

where coordinates are normalized to $[-1, 1]$, following the convention of grid_sample function (Paszke et al., 2019). In addition, $\phi_{\text{match}}$ predicts pixel-wise confidence maps $\hat{C}_s, \hat{C}_t \in [0, 1]^{H \times W}$ by adding one dimension, which provides reliability estimation for correspondences.

**Training Objectives.** The model is optimized through a progressive training recipe (Sec. 3.4) with: (i) *Supervised Loss ($L_2$ Loss)* from real data with sparse keypoints and synthetic data (Sec. 3.3) with dense correspondence labels (*i.e.*, grids); (ii) *Cycle-Consistency Loss* (Sec. 3.2) to refine matching and learn prediction uncertainty; and (iii) *Smoothness Loss* to mitigate aliasing artifacts (Sec. 3.5).

## 3.2 CYCLE-CONSISTENT TRAINING STRATEGY

Given the scarcity of large-scale real image pairs with dense annotations and the need of considering cross-image invisibility issues during prediction, we train the model to predict grids $\hat{G}_{s \to t}, \hat{G}_{t \to s}$ and confidences $\hat{C}_s, \hat{C}_t$ with a cycle-consistent strategy, including matching-reconstruction consistency and error-confidence correlation, as shown in Fig. 2.

**Matching-Reconstruction Consistency.** Let $\mathcal{W}(\cdot, \boldsymbol{G})$ be grid_sample function, we can obtain matching and reconstruction images:

$$\hat{I}_{s \to t} = \mathcal{W}(\boldsymbol{I}_s, \hat{\boldsymbol{G}}_{s \to t}), \qquad \hat{I}_{t \to s} = \mathcal{W}(\boldsymbol{I}_t, \hat{\boldsymbol{G}}_{t \to s}), \tag{3}$$

$$\hat{I}_{s \circlearrowleft} = \mathcal{W}(\hat{I}_{s \to t}, \hat{\boldsymbol{G}}_{t \to s}), \qquad \hat{I}_{t \circlearrowleft} = \mathcal{W}(\hat{I}_{t \to s}, \hat{\boldsymbol{G}}_{s \to t}). \tag{4}$$

We then use $\boldsymbol{M}_s, \boldsymbol{M}_t \in \{0, 1\}^{H \times W}$ as object masks (Ravi et al., 2024; Medeiros, 2024), and obtain the matching loss $\mathcal{L}_{\text{matching}}$ and reconstruction loss $\mathcal{L}_{\text{reconstruction}}$:

$$\mathcal{L}_{\text{matching}} = \boldsymbol{M}_t \odot \left\| E(\boldsymbol{I}_t) - E(\hat{I}_{s \to t}) \right\|_2 \odot \hat{\boldsymbol{C}}_t + \boldsymbol{M}_s \odot \left\| E(\boldsymbol{I}_s) - E(\hat{I}_{t \to s}) \right\|_2 \odot \hat{\boldsymbol{C}}_s, \tag{5}$$

$$\mathcal{L}_{\text{reconstruction}} = \boldsymbol{M}_s \odot \left\| \boldsymbol{I}_s - \hat{I}_{s \circlearrowleft} \right\|_2 \odot \hat{\boldsymbol{C}}_s + \boldsymbol{M}_t \odot \left\| \boldsymbol{I}_t - \hat{I}_{t \circlearrowleft} \right\|_2 \odot \hat{\boldsymbol{C}}_t, \tag{6}$$

where $E$ is the DINO feature extractor. Furthermore, we weight predictions using the confidence maps $\hat{C}_s, \hat{C}_t$ to account for prediction uncertainty during matching.

**Error-Confidence Correlation.** We require the predicted confidence to be inversely correlated with the reconstruction error. Define per-pixel errors

$$\boldsymbol{e}_s = \left\| \boldsymbol{I}_s - \hat{I}_{s \circlearrowleft} \right\|_2, \qquad \boldsymbol{e}_t = \left\| \boldsymbol{I}_t - \hat{I}_{t \circlearrowleft} \right\|_2, \tag{7}$$

$$\boldsymbol{e}^*_s = \frac{\boldsymbol{e}_s - \min(\boldsymbol{M}_s \odot \boldsymbol{e}_s)}{\max(\boldsymbol{M}_s \odot \boldsymbol{e}_s) - \min(\boldsymbol{M}_s \odot \boldsymbol{e}_s)}, \qquad \boldsymbol{e}^*_t = \frac{\boldsymbol{e}_t - \min(\boldsymbol{M}_t \odot \boldsymbol{e}_t)}{\max(\boldsymbol{M}_t \odot \boldsymbol{e}_t) - \min(\boldsymbol{M}_t \odot \boldsymbol{e}_t)}, \tag{8}$$

and the reference confidences $C_s = 1 - e^*{}_s$, $C_t = 1 - e^*{}_t$. We correlate the confidence and error as:

$$\mathcal{L}_{\text{uncertainty}} = \left\| M_s \odot (C_s - \hat{C}_s) \right\|_1 + \left\| M_t \odot (C_t - \hat{C}_t) \right\|_1 - \lambda_{\text{conf}} \cdot \left( \sum M_s \odot \hat{C}_s + \sum M_t \odot \hat{C}_t \right). \quad (9)$$

## 3.3 Synthetic Data with Dense Annotations

We generate paired data with dense annotations by integrating a text-to-3D model and a multi-condition image generation model: (i) 3D Asset Generation: We select 18 categories from SPair-71k (Min et al., 2019), expand textual descriptions for each instance via ChatGPT (OpenAI, 2025), and input these descriptions into Trellis (Xiang et al., 2025)'s text-to-3D model to produce 3D assets. (ii) Rendering (Ravi et al., 2020): Each 3D asset is rendered from multiple viewpoints to generate RGB images and depth maps. The index of the corresponding 3D point for each pixel is recorded as an index map (serving as pixel-wise ground-truth labels). (iii) Multi-Condition Image Generation: For each rendered image, we extract a Canny (Canny, 1986) map and combine it with the depth map and different textual descriptions to condition the FLUX (Labs, 2024), synthesizing RGB images with diverse textures. (iv) Paired Data Construction: For view-aligned pairs, transformations (*e.g.*, rotation, scaling) are applied to a canonical grid to generate warping grids (*i.e.*, $G_{s \to t}$, $G_{t \to s}$), followed by steps (i)-(iii) to produce paired images. For view-unaligned pairs, warping grids are first derived by matching index maps between different 3D assets, then steps (i)-(iii) are used to generate paired images. The overview of synthetic data is presented in Appendix A.7.

## 3.4 Progressive Training Recipe

During training, we optimize the model through a four-stage progressive strategy: (i) Synthetic Data Pretraining (Dense Supervision): Train exclusively on synthetic data with dense ground-truth annotations, employing only the $L_2$ loss and smoothness loss (Sec. 3.5). This stage aims to equip the model with VGGT's manifold-preserving mapping capabilities that generalize across different instances. (ii) Real Data Adaptation (Sparse Keypoint Supervision): Introduce real data with sparse ground-truth keypoints by adding a keypoint $L_2$ loss to the existing losses. This facilitates the transfer of dense mapping capabilities from synthetic to real-world domains. (iii) Matching Refinement (Matching and Reconstruction): Further optimize by incorporating the matching loss $\mathcal{L}_{\text{matching}}$ and reconstruction loss $\mathcal{L}_{\text{reconstruction}}$ to enhance matching precision. (iv) Uncertainty Learning (Confidence Prediction): Finally, integrate the uncertainty loss $\mathcal{L}_{\text{uncertainty}}$ to learn error-dependent confidence, enabling the model to predict prediction reliability. More details are presented in Appendix A.2

## 3.5 Aliasing Artifacts and Mitigation

The matching head predicts continuous coordinates for discrete pixels, which induces aliasing artifacts, manifesting as noticeable checkerboard patterns (See 6th column of Fig. 6) in the sampled images due to the inherent ambiguity (where slight coordinate shifts in either direction remain plausible). To mitigate this, we employ smoothness loss: constraining adjacent pixel coordinates to be spatially coherent by reshaping the grid map into a vector and enforcing similarity between each position and its adjacent neighbor.

## 4 Experiments

### 4.1 Experimental Settings

**Implementation Details.** We use a VGGT-based Transformer with $L=24$ blocks, where the first 4 blocks are fixed, and the remaining 20 blocks are duplicated from VGGT to initialize for a new semantic branch. Additionally, a DPT decoder is added as the semantic matching head, extracting features from blocks [4, 11, 17, 23]. The model is trained for 5 days using a single A6000 GPU. The evaluation is performed on SPair-71k (Min et al., 2019) and AP-10k (Yu et al., 2021) (intra-species (I.S.), cross-species (C.S.), and cross-family (C.F.)).

**Metrics.** Our quantitative evaluation consists of two aspects: (i) Dense Matching (**Synthetic Dense**): We conduct tests on synthetic data with ground-truth labels, warp the images based on the predicted correspondences, and compute the sum of squared errors between the warped images and the annotated images. (ii) Sparse Matching: We adhere to standard settings (Gupta et al., 2023;

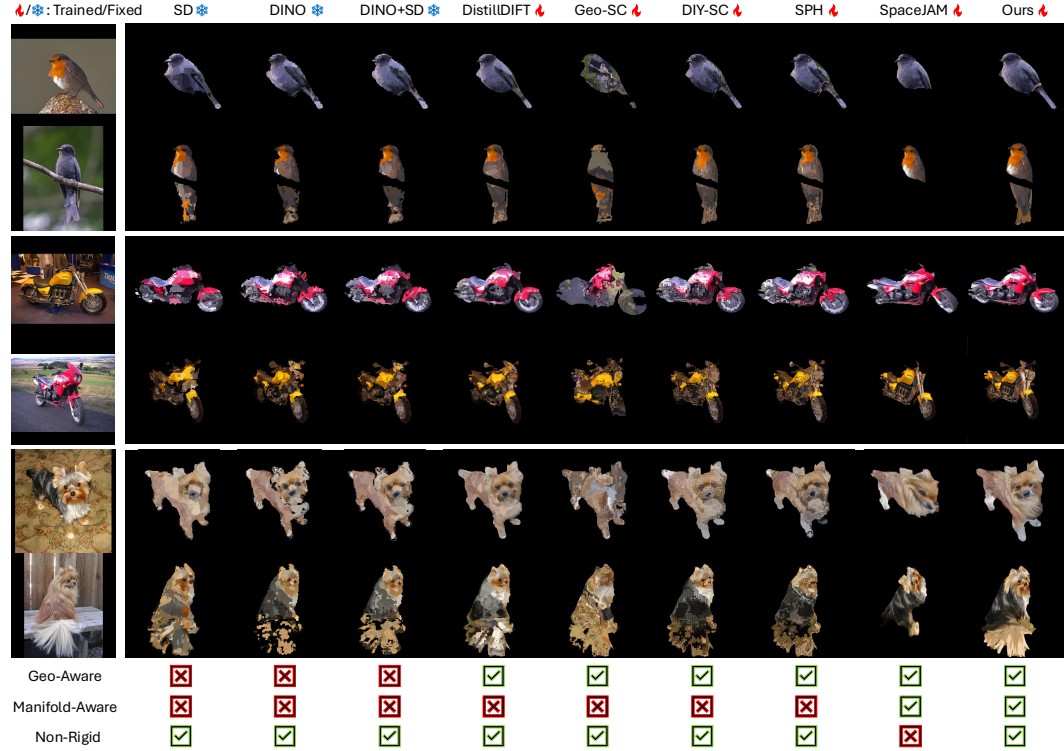

Figure 3: **Qualitative results of dense semantic matching compared with previous approaches.** We analyze them from four key dimensions: training requirements, geometric awareness, manifold preservation, and robustness to local non-rigidity. More results are provided in the Appendix A.5.

Table 1: **Quantitative results on SPair-71k (Min et al., 2019) and AP-10k (Yu et al., 2021).**

| Models | | SPair-71k(PCK ↑) | | | AP-10k (PCK@0.1 ↑) | | | Synthetic ↓ |
|---|---|---|---|---|---|---|---|---|
| | | 0.1 | 0.05 | 0.01 | I.S | C.S. | C.F. | Dense |
| SD + DINO | (Zhang et al., 2023a) | 59.9 | 44.7 | 7.9 | 62.9 | 59.3 | 48.3 | 0.20 |
| DistillDIFT* (U.S.) | (Fundel et al., 2025) | 60.8 | 45.4 | 8.0 | 65.8 | 64.2 | 56.1 | 0.16 |
| Geo-SC | (Zhang et al., 2024) | 65.4 | 49.1 | 9.9 | 68.7 | 64.6 | 52.7 | 0.14 |
| DistillDIFT* (W.S.) | (Fundel et al., 2025) | 65.3 | 49.8 | 8.9 | 66.9 | 64.7 | 58.0 | 0.15 |
| DIY-SC | (Dünkel et al., 2025) | 71.6 | 53.8 | 10.1 | 70.6 | 69.1 | 57.8 | 0.11 |
| SPH | (Mariotti et al., 2024) | 64.4 | 48.2 | 8.4 | 65.4 | 63.1 | 51.0 | 0.10 |
| SpaceJAM | (Barel et al., 2024) | 44.5 | 34.6 | 6.7 | 42.7 | 39.9 | 35.2 | 0.08 |
| Ours | | 76.8 | 57.2 | 14.5 | 72.8 | 70.1 | 60.5 | 0.08 |

Table 2: **Ablation study of VGGT backbone adaptation.**

| Models | | SPair-71k | Synthetic ↓ |
|---|---|---|---|
| Shared | DPT | PCK@0.1 ↑ | Dense |
| 18 | [4,11,17,23] | 53.0 | 0.23 |
| 12 | [4,11,17,23] | 62.7 | 0.19 |
| 6 | [4,11,17,23] | 72.4 | 0.09 |
| 4 | [4,11,17,23] | 76.8 | 0.08 |
| 4 | [3,10,16,22] | 75.6 | 0.10 |
| 4 | [2,9,15,21] | 74.7 | 0.12 |
| 4 | [1,8,14,20] | 72.9 | 0.10 |

Huang et al., 2022; Zhang et al., 2024; Xue et al., 2025; Mariotti et al., 2024; Min et al., 2019) and evaluate semantic correspondence performance using the Percentage of Correct Keypoints (**PCK**). PCK@$\alpha$ is defined as the ratio of correctly predicted matched keypoints that lie within a radius of $R = \alpha \cdot max(H, W)$ around their ground-truth points, with $H, W$ denoting the image size.

**Baseline Selection.** The selection of comparison approaches is based on two criteria: (i) Training Requirements: SD (Tang et al., 2023), DINO (Oquab et al., 2023), and SD+DINO (Zhang et al., 2023a) are training-free, whereas Geo-SC (Zhang et al., 2024), DIY-SC (Dünkel et al., 2025), SPH (Mariotti et al., 2024), and SpaceJAM(Barel et al., 2024) involve training. Notably, DistillDIFT (Fundel et al., 2025) ((U.S.) = No sparse keypoint tuning; (W.S.) = Sparse keypoint tuning applied) utilizes 3D synthetic data for training. (ii) Canonical Space Assumptions: SPH and SpaceJAM both adopt canonical space hypotheses. SPH explicitly requires instances of the same category to share a spherical canonical space, while SpaceJAM learns affine transformations projected onto a canonical space. The implementation of baselines are provided in Appendix A.3.

## 4.2 MATCHING EVALUATION

**Dense Matching.** Dense semantic correspondence prediction constitutes the core focus of our work. Unlike sparse keypoint matching, which focuses only on a few salient points, dense semantic match-

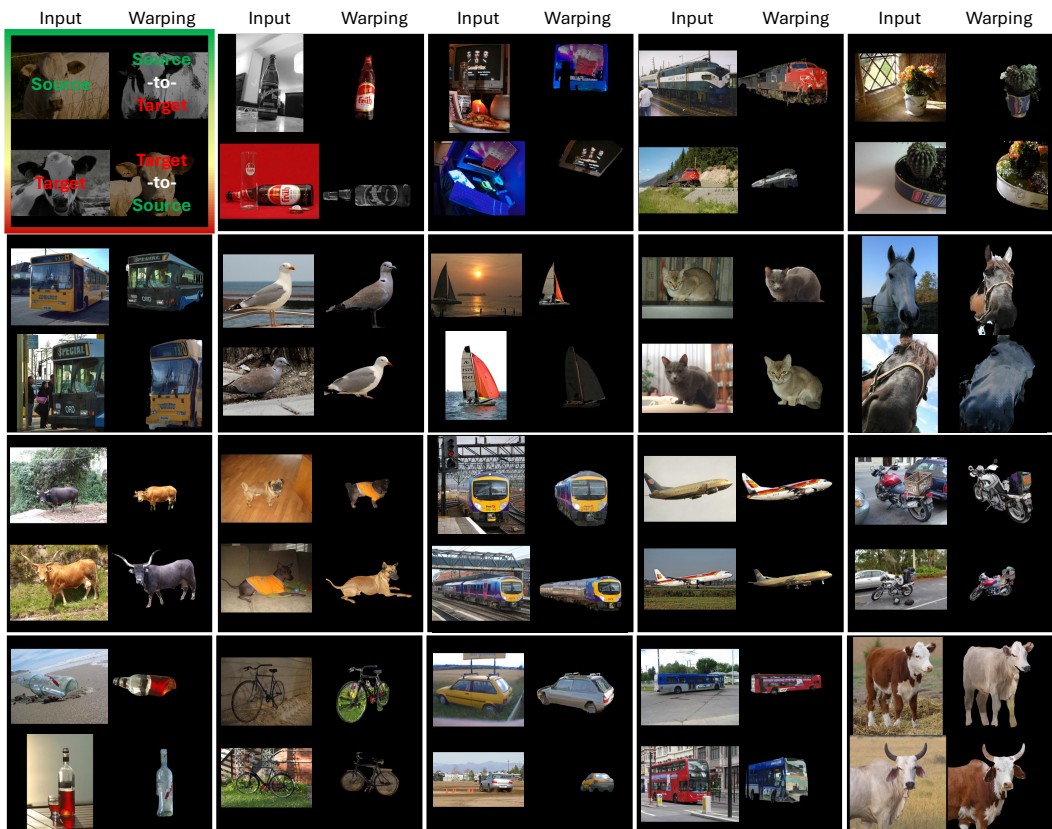

Figure 4: **More cases of dense semantic matching by our approach.** These results highlight our superior generalization across diverse object categories, viewpoints, and non-rigid deformations.

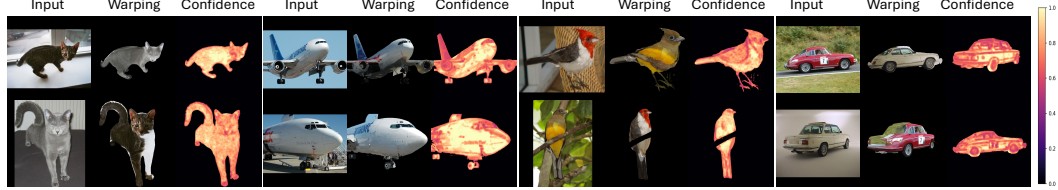

Figure 5: **More qualitative results of our approach on prediction reliability.** Confidence map prediction addresses the overlooked issue of cross-image invisibility, enhancing optimization via uncertainty calibration and offering reliability cues for downstream tasks.

ing requires pixel-wise correspondences across the entire image, demanding geometric awareness, manifold preservation, and robustness to local non-rigidity. We present qualitative comparisons with baseline approaches in Fig. 3, and analyze across four critical dimensions: (i) Training Requirements: While fine-tuned approaches consistently outperform zero-shot approaches, Geo-SC demonstrates limited effectiveness for dense matching despite improvements in keypoint matching. This stems from its keypoint-centric training objective that neglects dense correspondence. (ii) Geometric Awareness: Approaches incorporating geometric regularization, learning canonical spaces, or data augmentation exhibit superior performance in distinguishing symmetric and repetitive regions, a critical challenge for semantic matching. (iii) Manifold Preservation: With the exception of SpaceJAM and our approach, existing approaches fail to maintain underlying manifold structures during matching. This limitation compromises their utility for downstream tasks (*e.g.*, affordance learning) that require precise preservation of original surface geometries. (iv) Local Non-Rigidity: While SpaceJAM's global transformation learning preserves manifolds, its inability to handle non-rigid deformations restricts performance on objects with complex topologies. Only our approach satisfies all the requirements for dense semantic correspondence while addressing these fundamen-

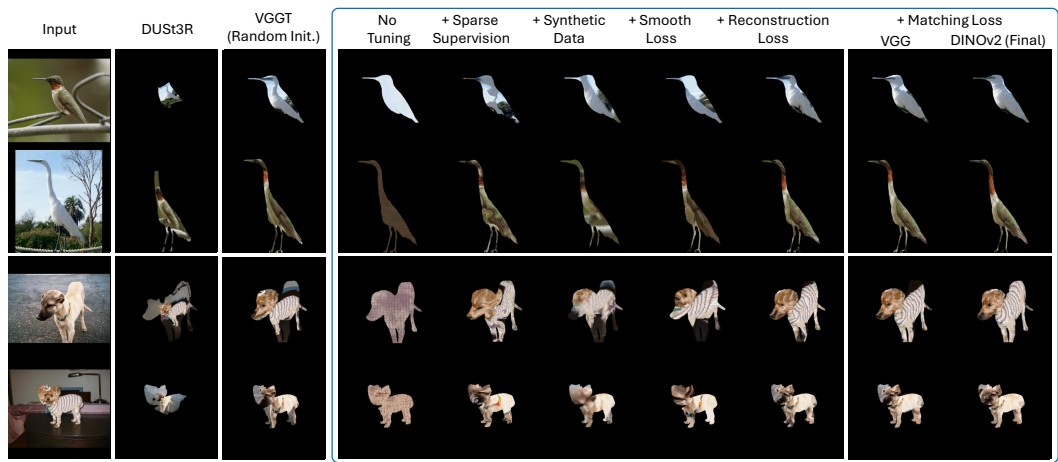

Figure 6: **Qualitative ablation results of the proposed approach.** Ablation study on two key aspects: (i) 3D reconstruction pretraining, showing that DUSt3R and randomly initialized VGGT are inferior to our approach in semantic matching accuracy and training efficiency; and (ii) training strategy design, where stage-wise incorporation of synthetic data, sparse supervision, smoothness, reconstruction, and matching losses highlights the effectiveness of our final training recipe.

Table 3: **Quantitative ablation evaluation of the proposed approach.** Abbreviations in the first row denote the ablation settings as shown in the first row of Fig. 6.

| | DUSt3R | VGGT(Rand.) | No Tuning | +Sparse | +Synthetic | +Smooth | +Recon. | +Match (VGG) | +Match (DINO) |
|---|---|---|---|---|---|---|---|---|---|
| SPair-71k (PCK@0.1) ↑ | 58.7 | 69.2 | 9.0 | 75.3 | 75.0 | 74.2 | 75.1 | 75.9 | **76.8** |
| Synthetic Dense ↓ | 0.15 | 0.13 | 0.49 | 0.19 | 0.12 | 0.11 | 0.11 | 0.10 | **0.08** |

tal limitations. Additional quantitative evaluations appear in Tab. 1, with extended qualitative results presented in Fig. 4 and Appendix A.5.

**Sparse Keypoint Matching.** Sparse keypoint matching evaluates the ability to match semantically salient regions. Although previous approaches (Fundel et al., 2025; Mariotti et al., 2024) attempted 3D synthetic data augmentation, they still struggle with geometry-ambiguous regions. In contrast, our approach surpasses previous approaches on this task, as shown in Tab. 1 and Appendix A.4.

**Matching Reliability.** We identify cross-image invisibility, where source pixels lack valid counterparts in the target image, as a critical yet overlooked factor that degrades matching reliability. Motivated by this, as shown in Fig. 5, we propose confidence map prediction, which not only introduces uncertainty calibration during matching learning to enhance optimization but also provides a reliability reference for downstream applications to selectively adopt high-confidence correspondences.

### 4.3 ABLATION STUDY

**Effectiveness of Each Key Design.** As shown in Fig. 6 and Tab. 3, we assess the effectiveness of our proposed approach through two key aspects: (i) 3D Reconstruction Pretraining Priors: We compare with two baselines: DUSt3R (Wang et al., 2024) and randomly initialized VGGT. While DUSt3R demonstrates manifold-preserving mapping capabilities, its reliance on inputs without the stronger representation power of 2D foundational model features (*e.g.*, DINO) limits its semantic matching performance under the same training settings. In contrast, randomly initialized VGGT achieves basic matching capability but requires significantly longer refinement time compared to our optimized approach. (ii) Training Strategy Analysis: Our analysis reveals that directly using VGGT's geometric correspondences without tuning fails for semantic matching. The introduction of sparse supervision enables basic matching but compromises manifold preservation. Incorporating synthetic data restores manifold structure yet introduces aliasing artifacts. While smoothness loss mitigates these artifacts, it reduces matching accuracy. Further addition of reconstruction loss improves performance but remains suboptimal. We compare VGG loss (Johnson et al., 2016) and DINO loss, ultimately selecting the latter as the optimal matching loss due to its superior fine-grained semantic matching accuracy. More detailed settings and results are provided in the Appendix A.6.

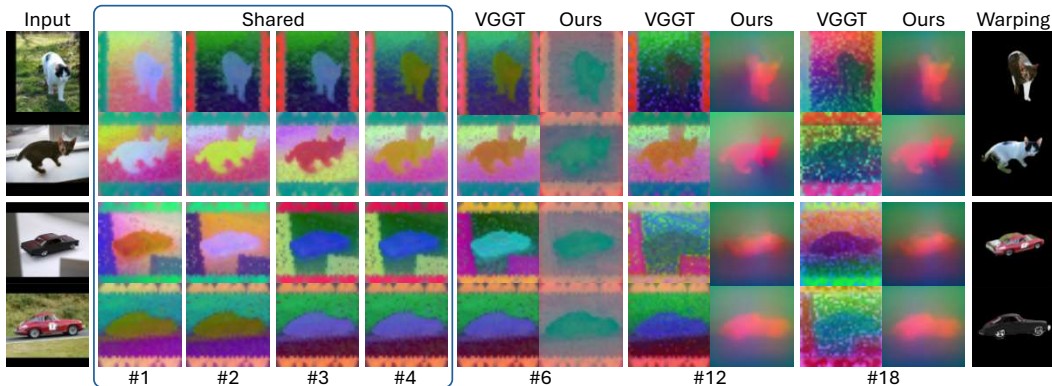

Figure 7: **Feature visualization of original VGGT backbone and our adapted semantic branch.** Unlike the original VGGT branch which yields noisy activations on cross-instance pairs, our fine-tuned semantic branch produces coherent and semantically aligned correspondences.

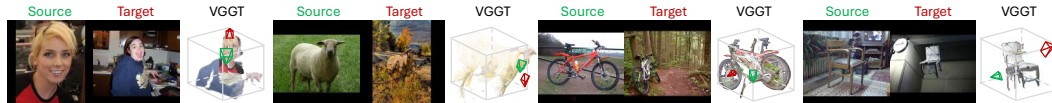

Figure 8: **More VGGT-predicted cases: coarse alignment & cross-instance semantic matching.**

**Backbone Block Selection and Feature Visualization.** We propose an effective architectural adaptation that augments VGGT with semantic matching while preserving its extensibility for broader downstream tasks. We conduct in-depth analyses about this adaptation on two key aspects: (i) Backbone Adaptation Analysis: Testing the blocks of transformer backbone, we ultimately adopt the first 4 blocks as shared components and retain features from blocks [4, 11, 17, 23] for DPT input. Quantitative validations are provided in Tab. 2. (ii) Feature Visualization: We employ Principal Component Analysis (PCA) to project features from different blocks into three RGB channels for visualization. Visual comparisons reveal that VGGT's original branch struggles with cross-instance image pairs, exhibiting weak feature correlations and noisy activations. However, our fine-tuned semantic branch demonstrates clear matching coherence and smooth feature distribution, as shown in Fig. 7.

**More VGGT Cases: Coarse Cross-Instance Alignment.** To further validate our initial motivation regarding VGGT's capability for coarse alignment across objects, we supplement additional examples in Fig. 8. While precise matching remains unachievable, semantically related regions exhibit approximate alignment.

## 5 CONCLUSION

Our work shows that integrating 3D reconstruction priors from VGGT into dense semantic matching provides a novel way to overcome geometric ambiguity, preserve manifold structures, and resolve cross-image invisibility. Methodologically, our approach integrates these key components: an architectural adaptation of VGGT that preserves its original ability and equip it with semantic matching capability; a cycle-consistent training strategy and a curated synthetic correspondence pipeline to alleviate annotation scarcity and cross-image invisibility; and a progressive training recipe with aliasing artifact mitigation that gradually transfers dense correspondence ability from synthetic domains to real-world data. Our approach not only surpasses existing techniques in dense semantic matching but also provides a practical extension of VGGT's paradigm applicable to diverse downstream tasks, highlighting the value of cross-task insights for advancing fundamental vision problems.

**Limitations and Future Work.** Although we propose a novel VGGT-based semantic matching approach, multi-view consistent semantic matching across scenes remains an open challenge. Additionally, to maintain compatibility with VGGT, we limit feature extraction to DINOv2. Future work could integrate additional features (*e.g.*, DINOv3 and Stable Diffusion) to enhance scalability, and further explore the use of the cycle-consistent strategy for self-supervised training on datasets with more diverse object categories. More results, failure cases, implementation details, and discussions are provided in the Appendix.

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

## A APPENDIX

We provide additional results to further demonstrate the effectiveness of our approach, along with detailed implementation settings to ensure reproducibility. In addition, we include extended discussions intended to inspire future research directions. For details, please refer to:

- A.1 Use of Large Language Models
- A.2 Implementation Details of Our Approach
- A.3 Baseline Implementations

### A.1 USE OF LARGE LANGUAGE MODELS

Large Language Models (LLMs) were used solely for minor grammar correction and stylistic polishing of the manuscript text. They were not involved in the design of the methodology, execution of experiments, analysis of results, or any other aspect of the scientific contribution.

### A.2 IMPLEMENTATION DETAILS OF OUR APPROACH

**Model Architecture.** To preserve the scalability of VGGT as a geometric foundation model and facilitate further research on downstream 3D-related tasks, we have made only minimal modifications to the original architecture. The overall pipeline remains consistent with that presented in the VGGT paper and its official codebase. Specifically, we duplicate the later transformer blocks to form a semantic branch for fine-tuning, and add a DPT layer to predict dense semantic correspondences. Dense Prediction Transformer (DPT) (Ranftl et al., 2021) is a transformer-based architecture for dense prediction tasks such as depth estimation and semantic segmentation . It leverages a Vision Transformer (ViT) as backbone to capture long-range dependencies through global self-attention, and employs a multi-stage decoder that fuses features from intermediate transformer layers to progressively restore spatial resolution. This hierarchical feature integration enables DPT to generate high-fidelity dense outputs with strong structural detail, making it well-suited for tasks requiring precise pixel-level prediction.

**Training Details.** Regarding training details, we emphasize a progressive training recipe. We initially attempted to apply all training techniques from scratch simultaneously, but this led to significant training instability, including issues such as gradient explosion and NaN values. Moreover, it became difficult to identify which specific component or loss term was responsible for the instability. To address this, we adopt a progressive training strategy. Training the full model on a single A6000 GPU takes five days in total: one day for training on synthetic data, one day for fine-tuning on real data, two days for refinement, and one additional day for uncertainty modeling. This step-by-step approach allows us to clearly monitor the capabilities acquired by the model at each stage and ensures stable and reliable training throughout the process. We use the Adam optimizer with a learning rate of $0.0001$. The weights for the different loss terms are set as follows: the dense supervision loss is $10,000$, the sparse supervision loss is $0.1$, the reconstruction loss is $100$, the matching loss is $2,000$, the smoothness loss is $1,000$, and the uncertainty loss is $0.01$. Additionally, $\lambda_{conf}$ is set to $0.1$. In practice, we split the later training epochs using synthetic and real data, with a ratio of 1:3. The key difference lies in the supervision signal: synthetic data is trained using the dense $L_2$ loss, while real data uses keypoint $L_2$ loss.

**More Discussions on DPT and Tracking-Based Matching.** In the original VGGT paper, the dense feature maps output by DPT are used for downstream tracking tasks based on CoTracker (Karaev et al., 2024). However, in our preliminary experiments, we find that these dense features without fine-tuning lack the ability to capture semantic correspondences. While fine-tuning could improve semantic alignment, integrating such fine-tuned features into a tracking paradigm like CoTracker would incur significant computational overhead. This is because CoTracker relies on a sparse set of pixels to query the feature map, whereas our task requires dense predictions. Joint training under this paradigm would thus be highly inefficient. Instead, we opt for direct grid prediction, which greatly improves training efficiency while maintaining dense correspondence accuracy.

**More Discussions on Bidirectional Correspondence Prediction.** An alternative design is to predict correspondence maps with respect to the first image as a reference. For example, in VGGT all frames are aligned to the first frame to estimate functional maps. We also experimented with this paradigm in the early exploration for predicting semantic correspondence maps, but observed

that training under this setting was inefficient and ultimately failed to match the performance of bidirectional prediction. Nevertheless, we do not dismiss the potential of this first-frame–referenced formulation, though extending it into a competitive paradigm requires further investigation.

## A.3 BASELINE IMPLEMENTATIONS

**SD: Stable Diffusion (DIFT).** DIFT (Tang et al., 2023) leverages the characteristics of Stable Diffusion, which are trained to denoise images by adding noise to clean images. The feature extraction process involves first adding noise to the input image to simulate the diffusion process, then feeding this noisy image into the trained U-Net of the diffusion model, alongside the current time step $t$, to extract feature maps. Specifically, a time step of $t = 261$ is used for semantic matching. The extracted feature maps are then used for pixel matching through nearest neighbor search using cosine distance. **Project: https://github.com/Tsingularity/dift.**

**DINO.** DINO (Oquab et al., 2023) is a self-supervised learning framework designed to learn visual representations without labeled data. It employs a contrastive learning approach, minimizing the distance between augmented views of the same image while maximizing the distance between views of different images. For semantic matching, DINO extracts features from images using a Vision Transformer (ViT), capturing various levels of abstraction. To perform semantic matching, DINO uses a nearest neighbor approach, comparing the feature representation of a query image with those of target images, employing cosine distance as the similarity metric. **Project: https://github.com/facebookresearch/dinov2.**

**DINO+SD.** Zhang et al. (2023a) explores the integration of Stable Diffusion (SD) features with DINOv2 features to enhance semantic matching between images. It begins with feature extraction from SD by adding noise to the input image and performing a denoising step using the latent code, capturing spatial layouts but occasionally lacking in semantic accuracy. DINOv2 features, extracted using ViT, provide sparse yet precise matches, complementing the rich spatial information from SD. Semantic correspondence is established through zero-shot evaluation, employing a nearest neighbor search on the fused features to find the closest vectors between images. **Project: https://github.com/Junyi42/sd-dino.**

**DistillDIFT.** DistillDIFT (Fundel et al., 2025) extract features from various stages of the denoising process, focusing on capturing both low-level and high-level semantic information. To improve the robustness of these features, a distillation framework is employed, where a teacher model (the diffusion model) guides a student model in learning to generate more accurate and semantically meaningful features. This framework includes a contrastive loss that encourages the student to produce features that are invariant to different augmentations of the same input. The resulting distilled features are then used for semantic correspondence tasks, allowing for effective matching between images. The experiments demonstrate that this distillation method significantly improves performance on benchmark datasets, showcasing the potential of diffusion features in zero-shot semantic correspondence scenarios. **Project: https://github.com/CompVis/distilldift.**

**Geo-SC.** Geo-SC (Zhang et al., 2024) introduces a method for establishing semantic correspondences by incorporating geometric awareness into the matching process. Specifically, the method employs a geometry-aware attention mechanism that helps the model focus on relevant spatial relationships between keypoints. This attention mechanism is guided by geometric information derived from the images, allowing the model to differentiate between left and right structures, which is crucial for many semantic tasks. The framework also includes a novel loss function that encourages consistency in correspondence while exploiting geometric priors. **Project: https://github.com/Junyi42/GeoAware-SC.**

**DIY-SC.** DIY-SC (Dünkel et al., 2025) presents an approach to learn semantic correspondences from pseudo-labels generated with the aid of foundation model features and geometric priors. The framework produces coarse matches via nearest-neighbor feature matching and then filters them using relaxed cycle consistency, chained image pairs, and spherical prototype rejection to obtain high-quality pseudo-labels. These pseudo-labels are used to train a lightweight adapter with both sparse and dense losses, enabling the model to refine feature representations for robust correspondence learning. **Project: https://genintel.github.io/DIY-SC.**

**SPH.** SPH (Mariotti et al., 2024) proposes a method that transforms images into spherical representations, allowing for better alignment of features across different viewpoints. By incorporating geometric information related to the viewpoint, the method effectively captures the spatial relationships between objects, facilitating more accurate semantic matching. The framework employs a viewpoint-guided attention mechanism that selectively focuses on relevant areas of the spherical maps, improving the model's ability to discern correspondences despite variations in perspective. **Project: https://github.com/VICO-UoE/SphericalMaps.**

**SpaceJAM.** SpaceJAM (Barel et al., 2024) is a lightweight method that performs joint alignment across multiple images by leveraging a direct optimization framework. This approach allows for rapid computation of alignment parameters without the overhead typically associated with regularization, making it suitable for real-time applications. By employing a novel feature extraction strategy, the method effectively captures keypoints and descriptors that are robust to variations in image content and perspective. **Project: https://github.com/BGU-CS-VIL/SpaceJAM.**

### A.4    MORE SPARSE MATCHING RESULTS

We first focus on semantic matching at sparse keypoints, which is the limitation of previous work, and we thoroughly evaluate this core task. We visualize representative results to qualitatively illustrate geometry-aware keypoint matching, as shown in Fig. 9. In addition, we summarize per-category performance, as reported in Tab. 4.

### A.5    MORE DENSE MATCHING RESULTS

Compared with previous approaches, ours emphasizes the importance of dense semantic matching and achieves geometry-aware, manifold-preserving matching from taming 3D reconstruction priors. We present extensive dense semantic matching results, as shown in Fig. 10, Fig. 11, Fig. 12, Fig. 13, Fig. 14, Fig. 15, Fig. 16, Fig. 17, Fig. 18, Fig. 19, Fig. 20, Fig. 21, Fig. 22, Fig. 23, and Fig. 24.

### A.6    ABLATION STUDY SETTING AND MORE RESULTS

The ablation study settings are: (i) "DUSt3R": One day for training on synthetic data, one day for fine-tuning on real data, and two days for refinement, and one additional day for uncertainty modeling. (ii) "VGGT (Random Init.)": Two day for training on synthetic data, two day for fine-tuning on real data, three days for refinement, and one additional day for uncertainty modeling. (iii) "No Tuning": Use the original checkpoint of VGGT. (iv) "+Sparse Supervision": One day for training on real image dataset with sparse keypoint annotation. (v) "+Synthetic Data": One day for training on synthetic data and one day for fine-tuning on real data without smoothness loss. (vi) "+Smooth Loss": One day for training on synthetic data and one day for fine-tuning on real data with smoothness loss. (vii) "+Reconstruction Loss": One day for training on synthetic data, one day for fine-tuning on real data, two days for refinement with only reconstruction loss. (viii) "+Matching Loss VGG": One day for training on synthetic data, one day for fine-tuning on real data, two days for refinement with reconstruction loss and VGG matching loss. (ix) "+Matching Loss DINO (Final)": One day for training on synthetic data, one day for fine-tuning on real data, two days for refinement with reconstruction loss and DINO matching loss, and one additional day for uncertainty modeling. We further present more qualitative ablation results to provide stronger evidence for the effectiveness of the proposed approach, as shown in Fig. 25 and Fig. 26.

### A.7    OVERVIEW OF SYNTHETIC DATA

We sample a set of image pairs from synthetic datasets. As shown in Fig. 27, these data exhibit substantial diversity in category, viewpoint, and invisibility. However, such synthetic data remain constrained by current image-generation capabilities and still fall short of real data in photo-realism.

### A.8    FAILURE CASES AND ANALYSIS

Although our approach provides a novel perspective for dense semantic matching, several limitations remain due to the relatively small dataset size and the limited diversity of object categories covered during training. In extensive testing, we identified three notable types of failure cases: (i) Precise

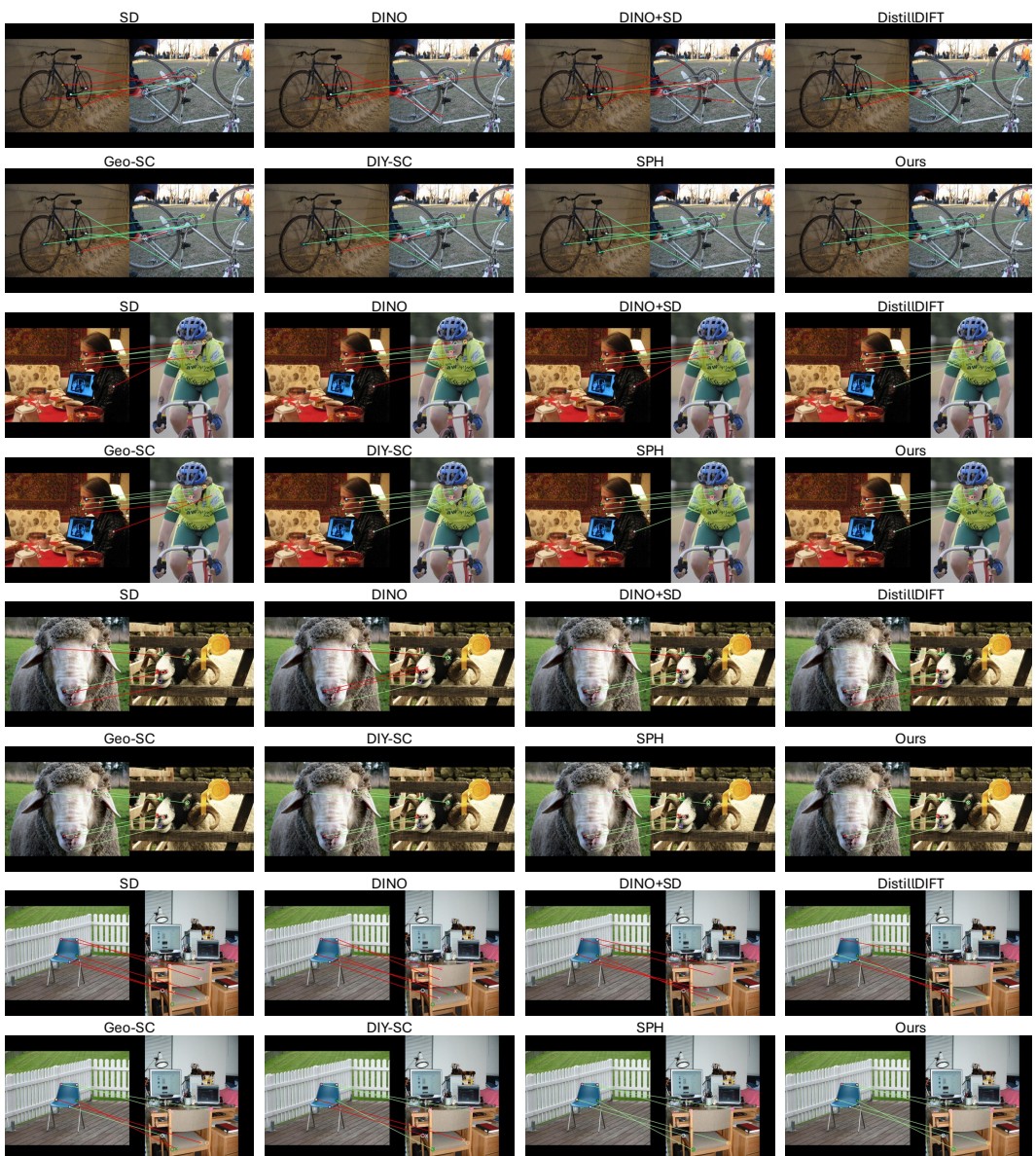

Figure 9: **Comparisons of sparse keypoint matching.**

Table 4: **The quantitative results of per-category PCK@0.1 scores on SPair-71k.**

| | | ✈ | 🚲 | 🐦 | 🚤 | 🍾 | 🚌 | 🚗 | 🐱 | 🪑 | 🐄 | 🐕 | 🐎 | 🏍 | 🚶 | 🪴 | 🐑 | 🚆 | 🖥 | ALL |
|---|---|---|---|---|---|---|---|---|---|---|---|---|---|---|---|---|---|---|---|---|
| DINO | (Oquab et al., 2023) | 72.7 | 62.0 | 85.2 | 41.3 | 40.4 | 52.3 | 51.5 | 71.1 | 36.2 | 67.1 | 64.6 | 67.6 | 61.0 | 68.2 | 30.7 | 62.0 | 54.3 | 24.2 | 55.6 |
| SD (DIFT) | (Tang et al., 2023) | 63.5 | 54.5 | 80.8 | 34.5 | 46.2 | 52.7 | 48.3 | 77.7 | 39.0 | 76.0 | 54.9 | 61.3 | 53.3 | 46.0 | 57.8 | 57.1 | 71.1 | 63.4 | 57.7 |
| SD + DINO | (Zhang et al., 2023a) | 73.0 | 64.1 | 86.4 | 40.7 | 52.9 | 55.0 | 53.8 | 78.6 | 45.5 | 77.3 | 64.7 | 69.7 | 63.3 | 69.2 | 58.4 | 67.6 | 66.2 | 53.5 | 64.0 |
| DistillDIFT* (U.S.) | (Fundel et al., 2025) | 74.6 | 60.4 | 88.7 | 42.5 | 53.5 | 55.0 | 54.6 | 80.8 | 42.7 | 78.6 | 72.0 | 71.4 | 62.2 | 70.7 | 53.1 | 68.6 | 65.2 | 61.6 | 65.1 |
| DistillDIFT* (W.S.) | (Fundel et al., 2025) | 78.2 | 63.8 | 90.1 | 45.0 | 54.6 | 68.0 | 63.7 | 83.2 | 49.3 | 82.6 | 74.5 | 73.8 | 63.5 | 72.0 | 56.2 | 71.0 | 86.2 | 66.5 | 70.6 |
| Geo-SC | (Zhang et al., 2024) | 78.0 | 66.4 | 90.2 | 44.5 | 60.1 | 66.6 | 60.8 | 82.7 | 53.2 | 82.3 | 69.5 | 75.1 | 66.1 | 71.7 | 58.9 | 71.6 | 83.8 | 55.5 | 69.6 |
| DIY-SC | (Dünkel et al., 2025) | 77.2 | 69.1 | 90.8 | 54.2 | 57.9 | 83.7 | 77.5 | 86.5 | 53.1 | 86.7 | 73.1 | 78.5 | 72.5 | 74.0 | 73.5 | 76.0 | 77.2 | 69.5 | 74.4 |
| SPH | (Mariotti et al., 2024) | 75.3 | 63.8 | 87.7 | 48.2 | 50.9 | 74.9 | 71.1 | 81.7 | 47.3 | 81.6 | 66.9 | 73.1 | 65.4 | 61.8 | 55.5 | 70.2 | 75.0 | 58.5 | 67.8 |
| SpaceJAM | (Barel et al., 2024) | 53.6 | 53.4 | 45.4 | 47.5 | 71.0 | 54.0 | 46.0 | 66.0 | 25.8 | 48.6 | 28.5 | 47.6 | 54.0 | 50.7 | 34.0 | 9.0 | 71.8 | 15.4 | 45.7 |
| Ours | | 79.1 | 70.2 | 90.9 | 56.8 | 72.4 | 85.1 | 79.2 | 90.8 | 58.1 | 88.4 | 76.1 | 79.3 | 75.2 | 75.6 | 78.2 | 79.7 | 88.1 | 71.4 | 77.5 |

Structural Matching: For objects with highly intricate structures, such as the bicycle gears and pedals shown in Fig. 28 (a), our current model struggles to achieve high accuracy without paired training data and more advanced optimization strategies. (ii) Complex Non-Rigid Matching: For objects or animals with significant non-rigid deformations or complex motion patterns, such as cats in Fig. 28 (b) with frequently extreme poses, the model performs less reliably when learning mappings for complex regions like the legs, compared to more stable regions such as the face. (iii) Complete Reversal Problem: For perfectly axis-symmetric objects, such as the two sides of a cow's body

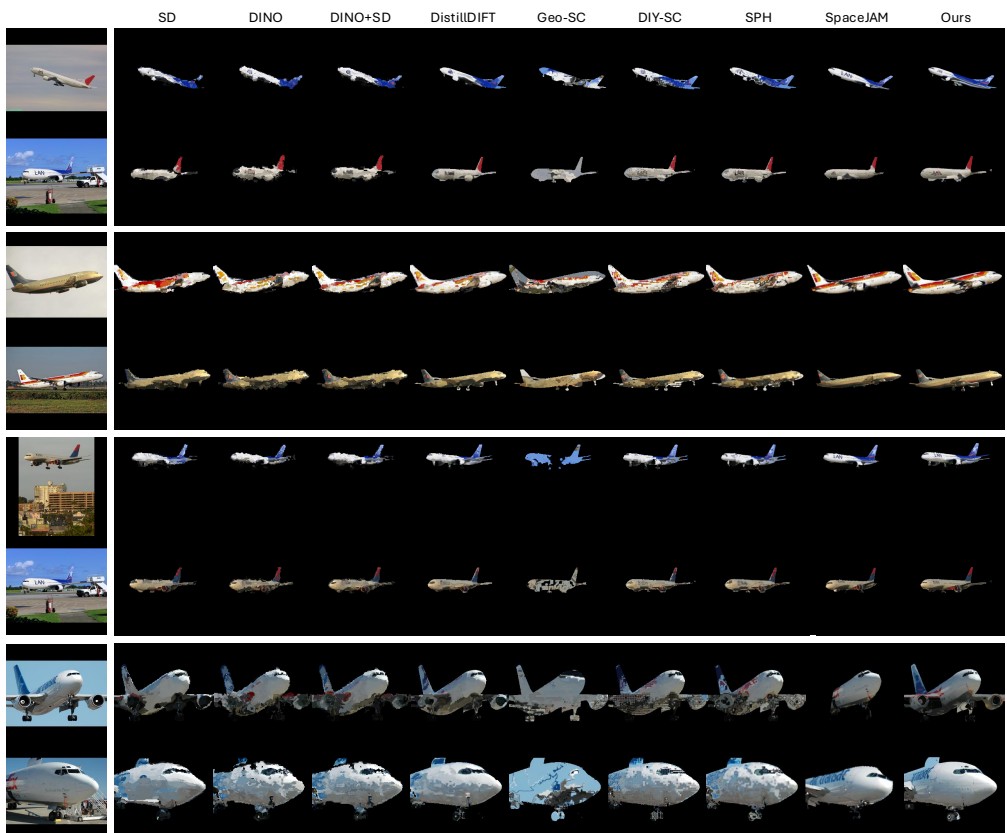

Figure 10: **Comparisons of dense semantic matching (Aeroplane).**

in Fig. 28 (c), our current model design and optimization objectives provide no explicit incentive to discourage flipping. As a result, the model often "adapts" by reversing correspondences to minimize loss.

To address these failure cases, future work could expand the dataset in both scale and category diversity, and explore training on larger unannotated collections using self-supervised strategies such as our cycle-consistent training strategy to enhance robustness. With greater computational resources and richer data augmentation, our approach can be further scaled to handle more challenging cases, including humans and animals with complex structures, non-rigid deformations, and diverse poses. We also expect it to inspire follow-up research that leverages correspondence prediction as a foundation for downstream applications such as motion transfer (Yang et al., 2024; Rekik et al., 2024), virtual try-on (Kim et al., 2024), 3D morphing (Yang et al., 2025), and affordance transfer (Lai et al., 2021).

### A.9 DISCUSSIONS OF NEW DINO MODEL: DINOV3

Recently, the new version of DINO, DINOv3(Siméoni et al., 2025), has attracted considerable attention, particularly with the release of its 7B-parameter model. The DINOv3 family has achieved notable progress in tasks such as image classification and depth estimation. Motivated by this, we conducted semantic matching experiments using Meta's official feature extraction code (**Project: https://github.com/facebookresearch/dinov3.**). Interestingly, two observations emerged that merit further investigation: (i) DINOv3 performs poorly in cross-instance semantic matching; (ii) increasing the model size leads to degraded semantic correspondence. These findings suggest that for our dense semantic matching task, leveraging the pretrained strengths of VGGT is more efficient and effective than training directly with DINOv3 from scratch. Nevertheless, the potential demonstrated by DINOv3 highlights promising directions for exploring its applicability to dense semantic matching in future work.

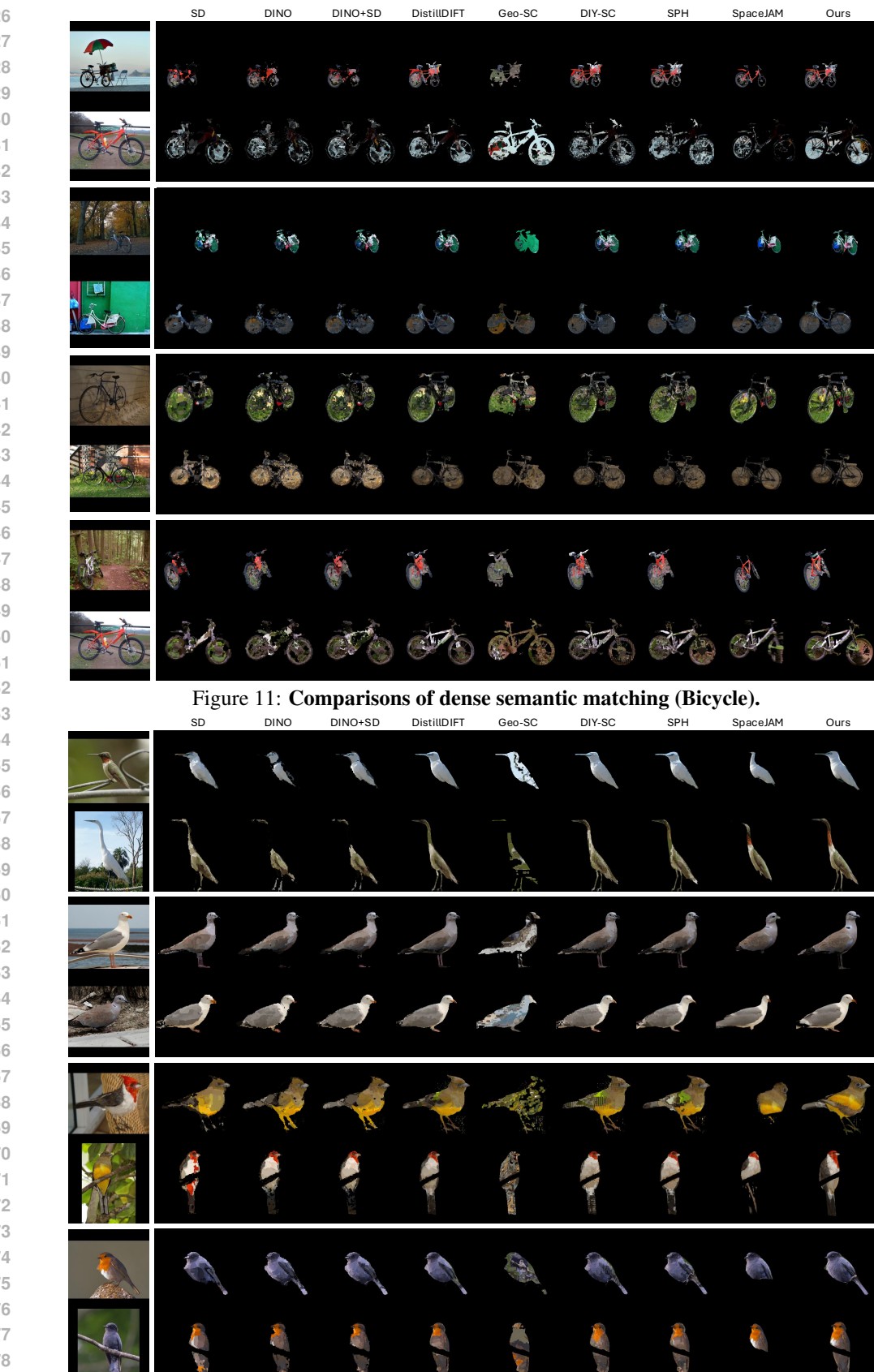

Figure 11: **Comparisons of dense semantic matching (Bicycle).**

Figure 12: **Comparisons of dense semantic matching (Bird).**

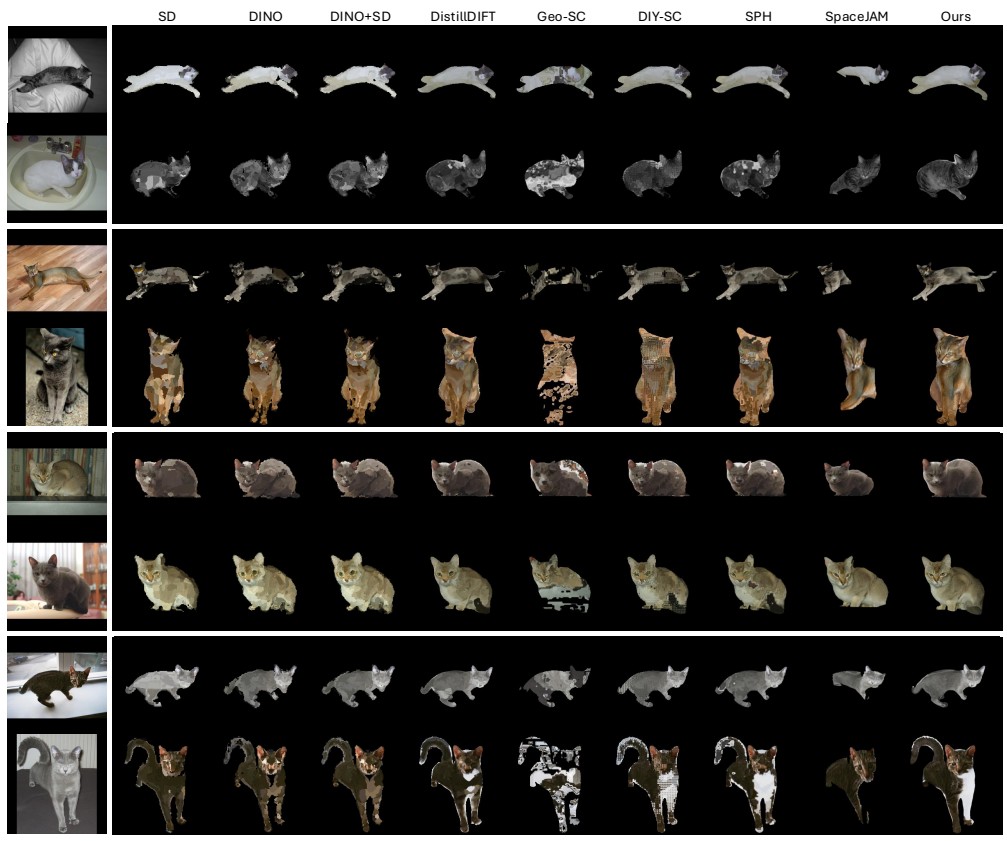

Figure 13: **Comparisons of dense semantic matching (Cat).**

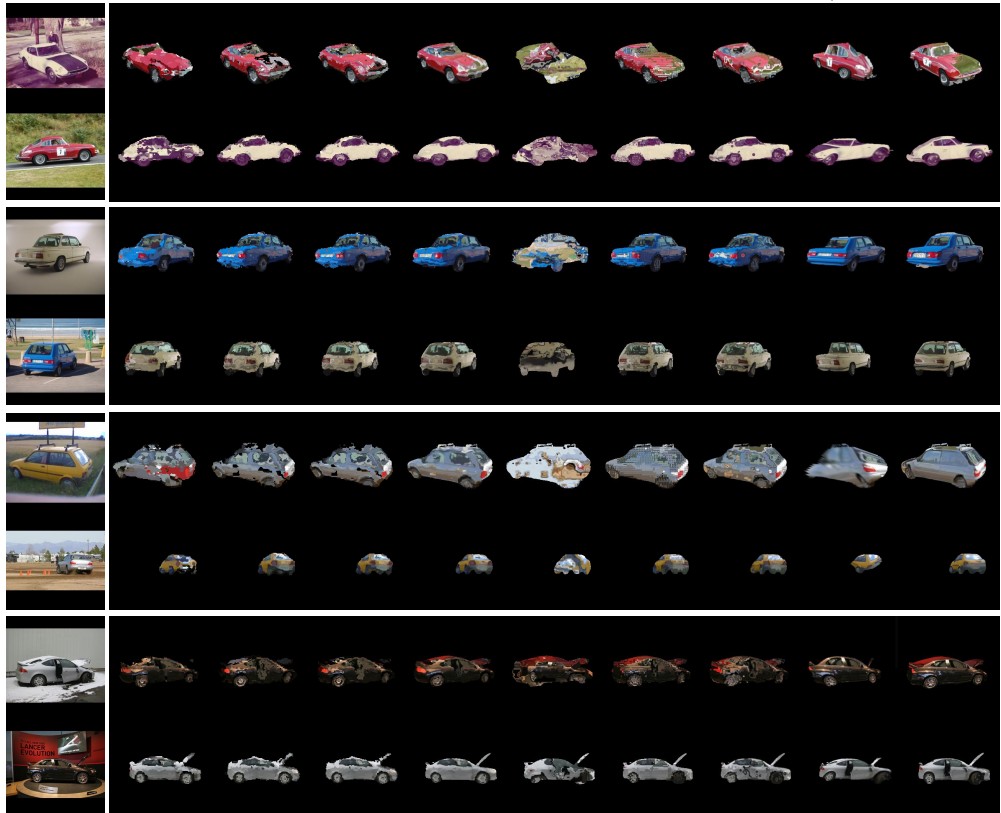

Figure 14: **Comparisons of dense semantic matching (Car).**

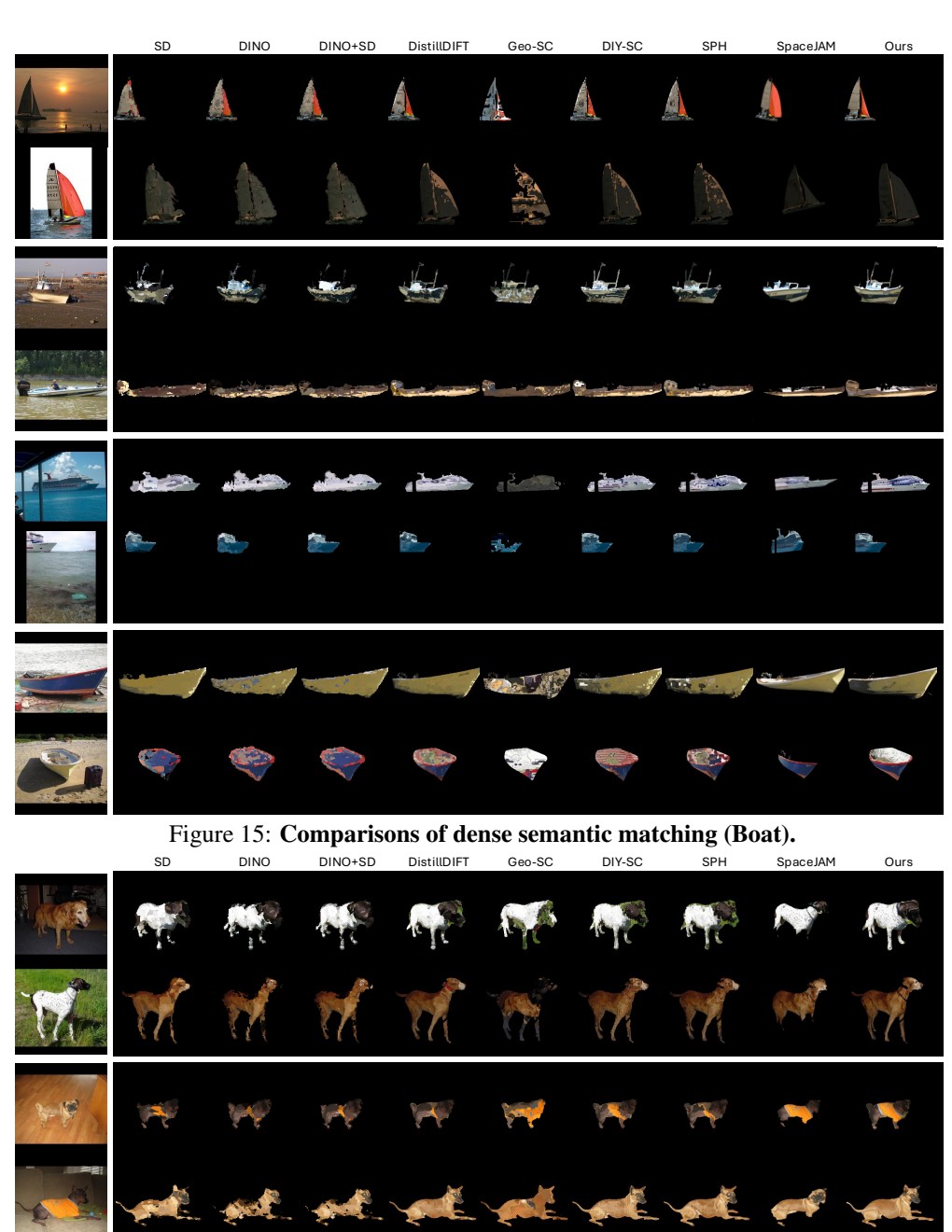

Figure 15: **Comparisons of dense semantic matching (Boat).**

Figure 16: **Comparisons of dense semantic matching (Dog).**

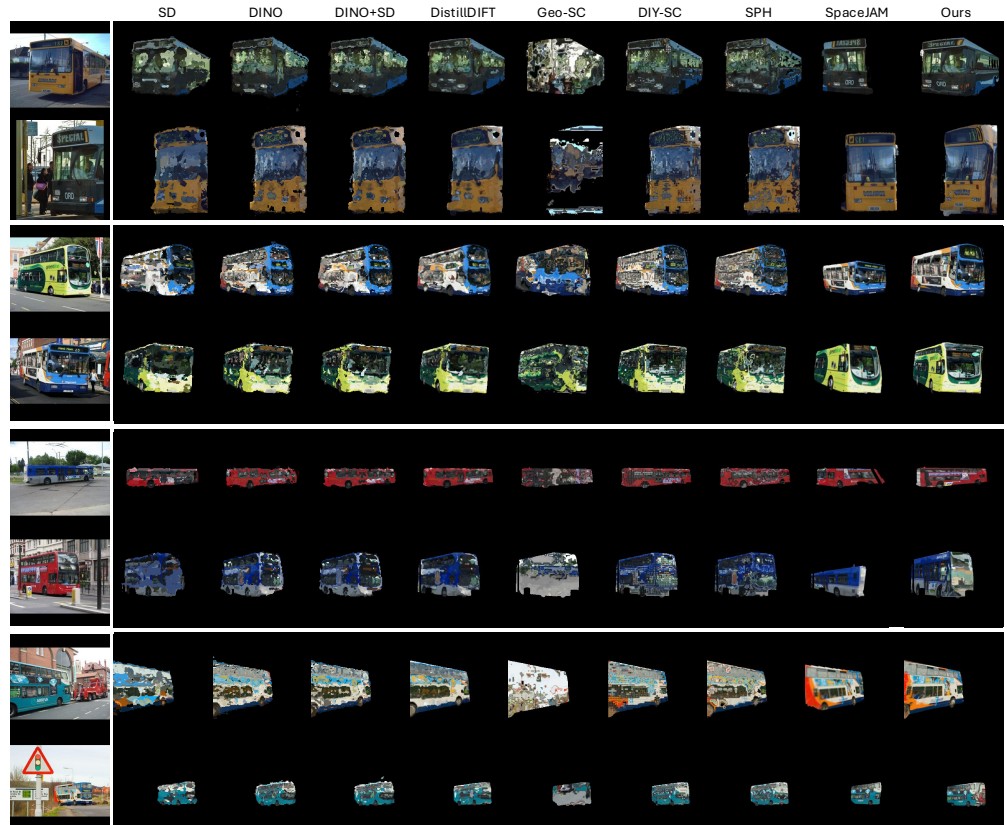

Figure 17: **Comparisons of dense semantic matching (Bus).**

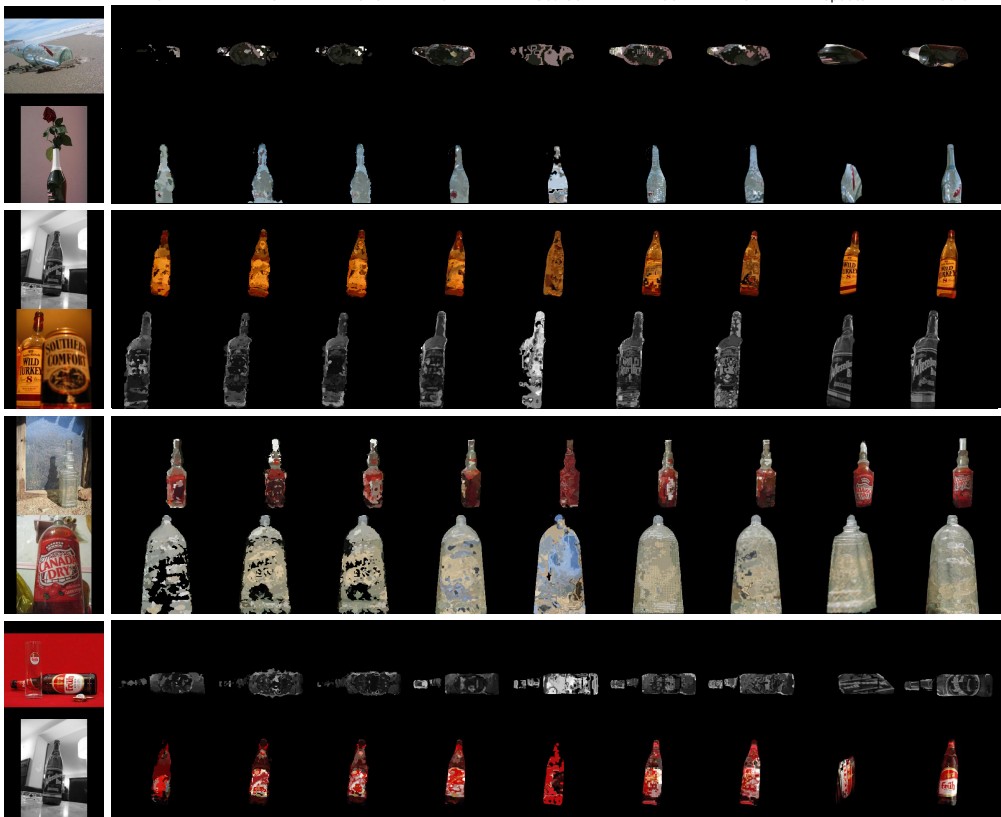

Figure 18: **Comparisons of dense semantic matching (Bottle).**

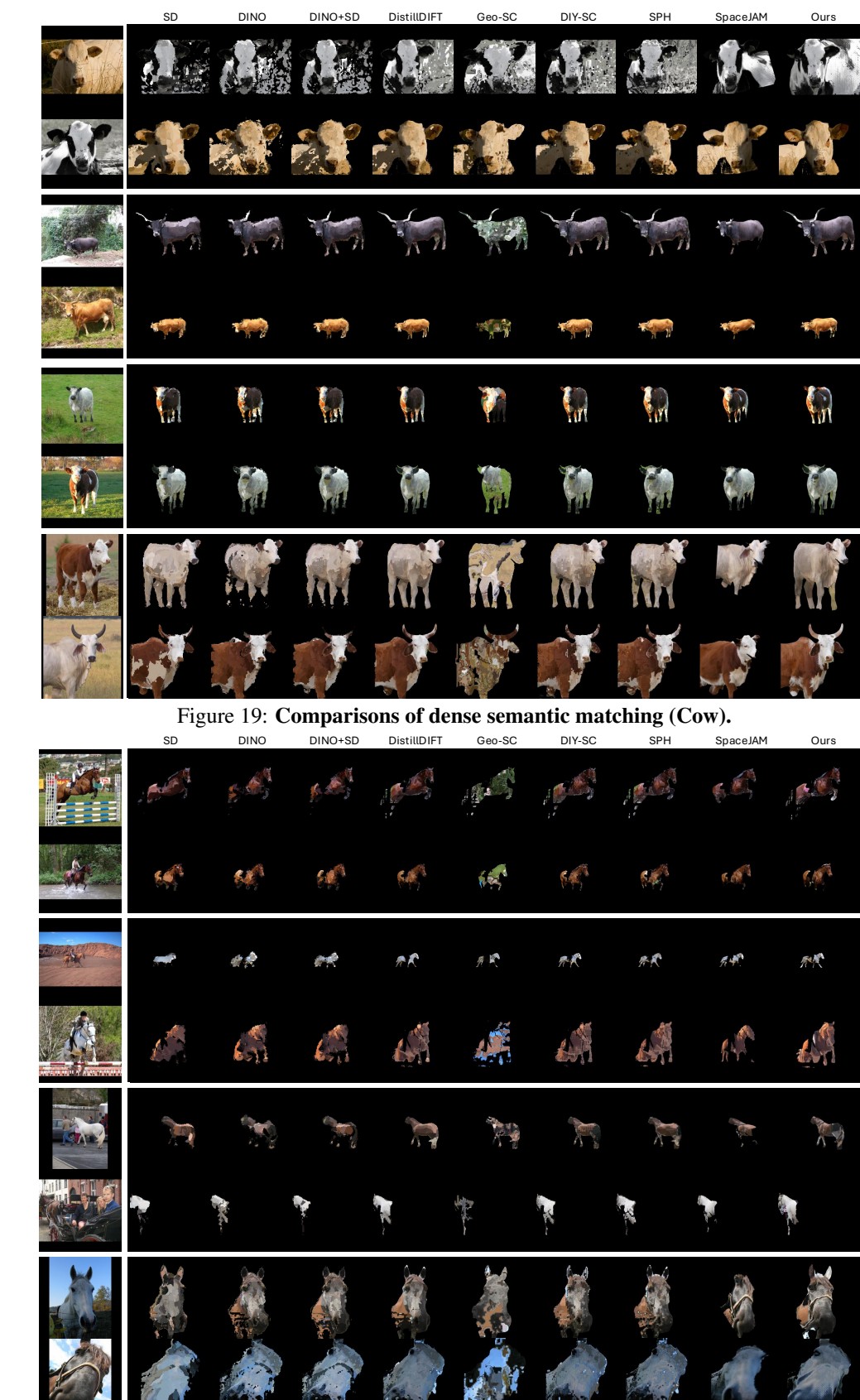

Figure 19: **Comparisons of dense semantic matching (Cow).**

Figure 20: **Comparisons of dense semantic matching (Horse).**

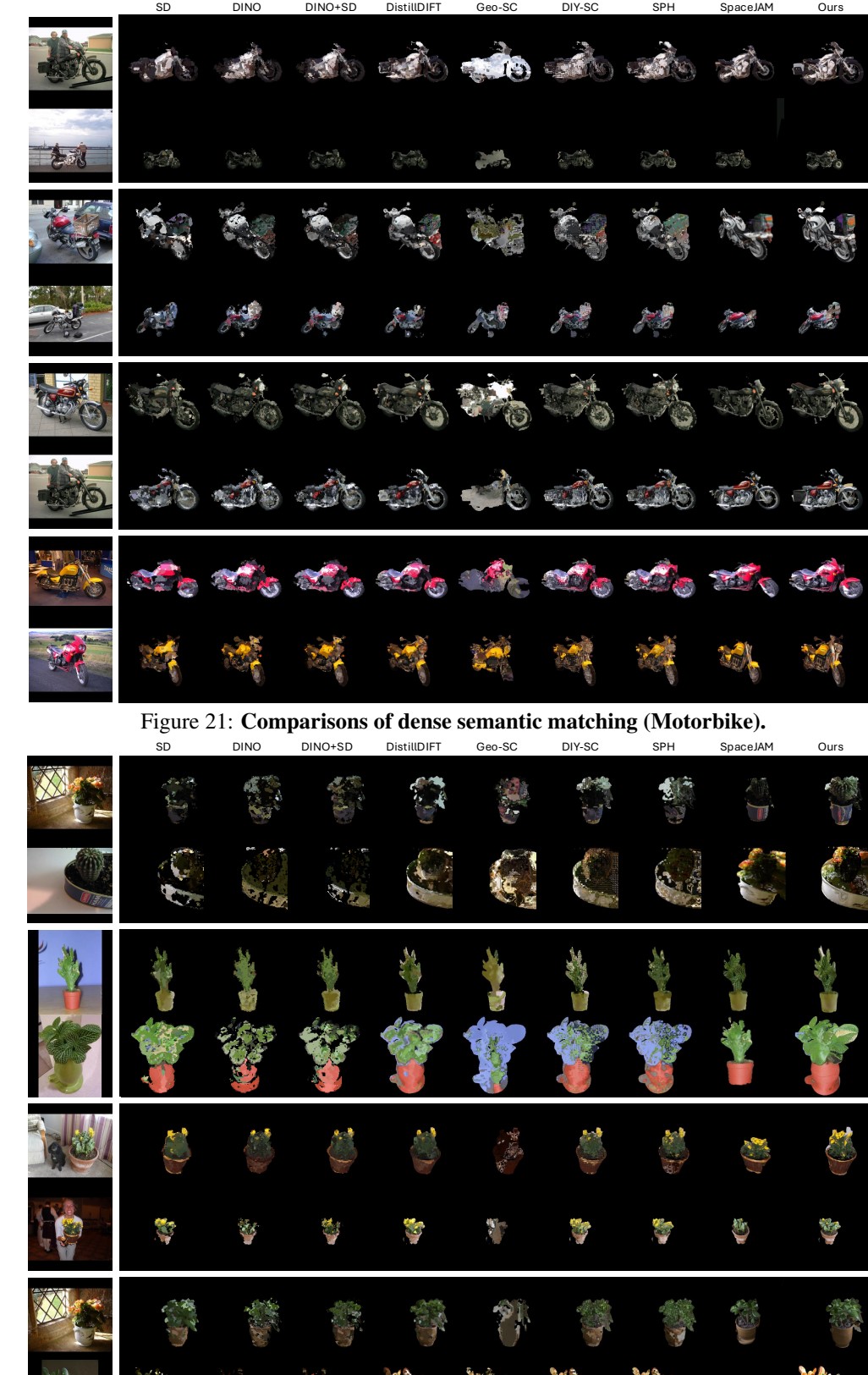

Figure 21: **Comparisons of dense semantic matching (Motorbike).**

Figure 22: **Comparisons of dense semantic matching (Potted Plant).**

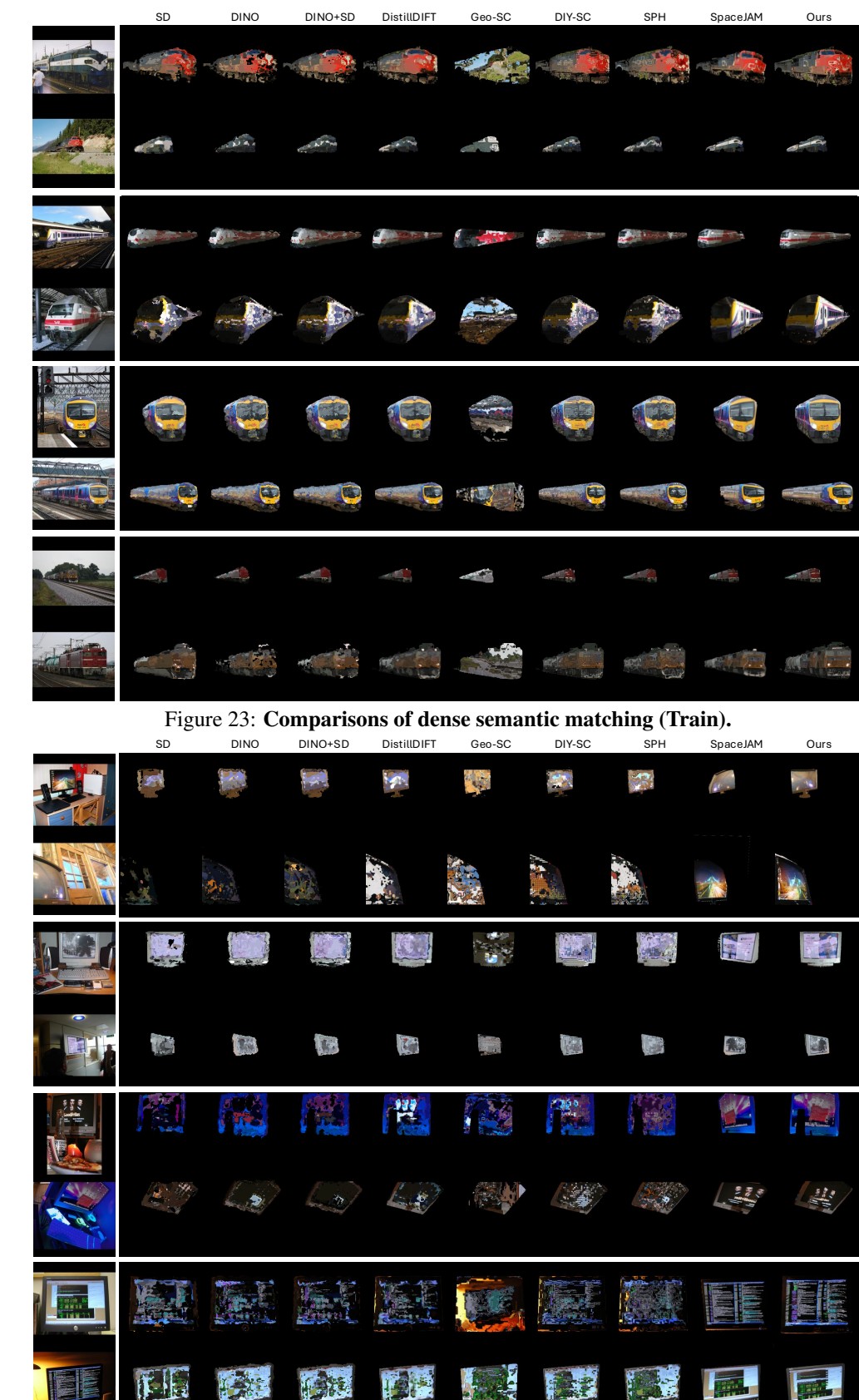

Figure 23: **Comparisons of dense semantic matching (Train).**

Figure 24: **Comparisons of dense semantic matching (Tvmonitor).**

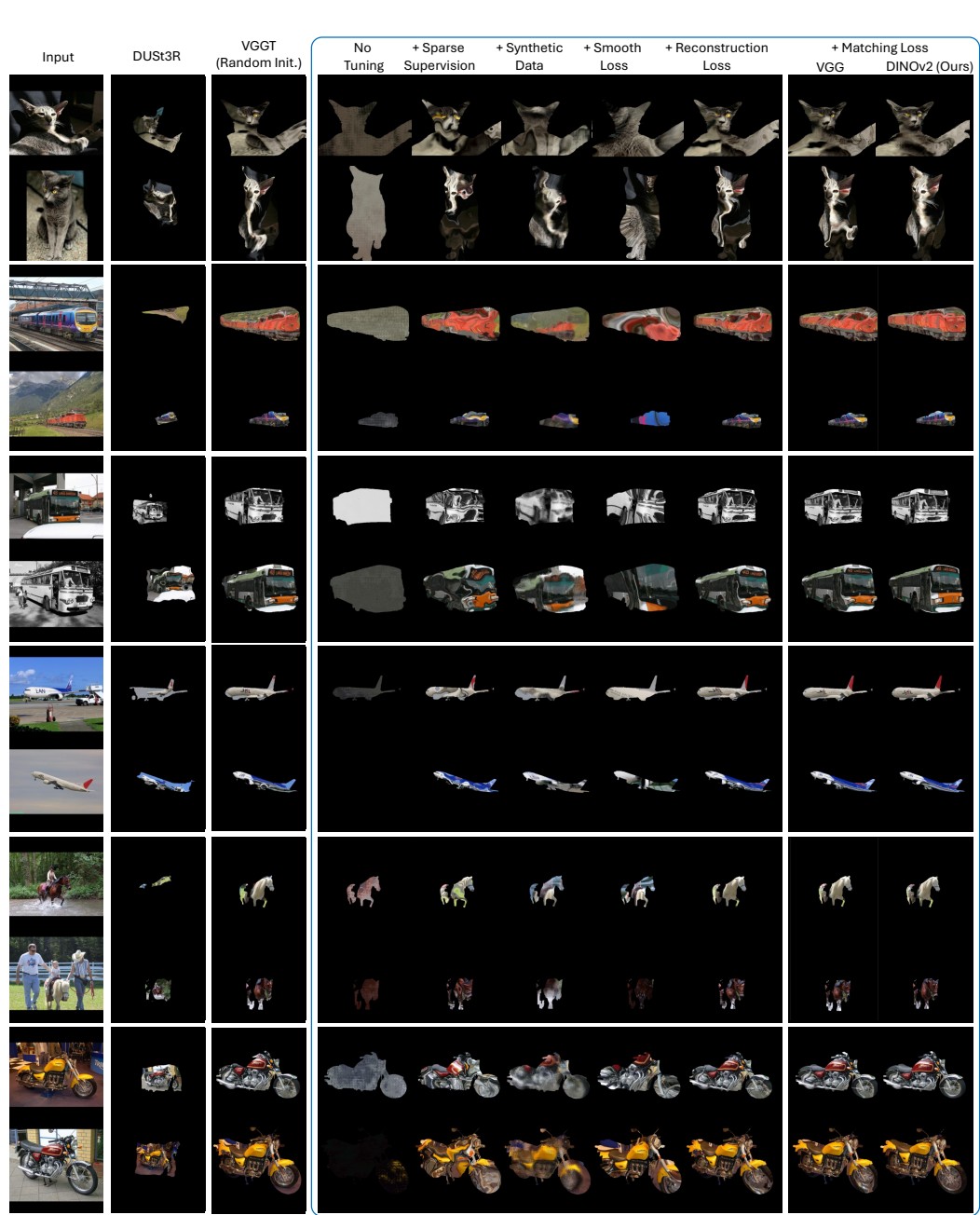

Figure 25: **More qualitative ablation results of the proposed approach (1).**

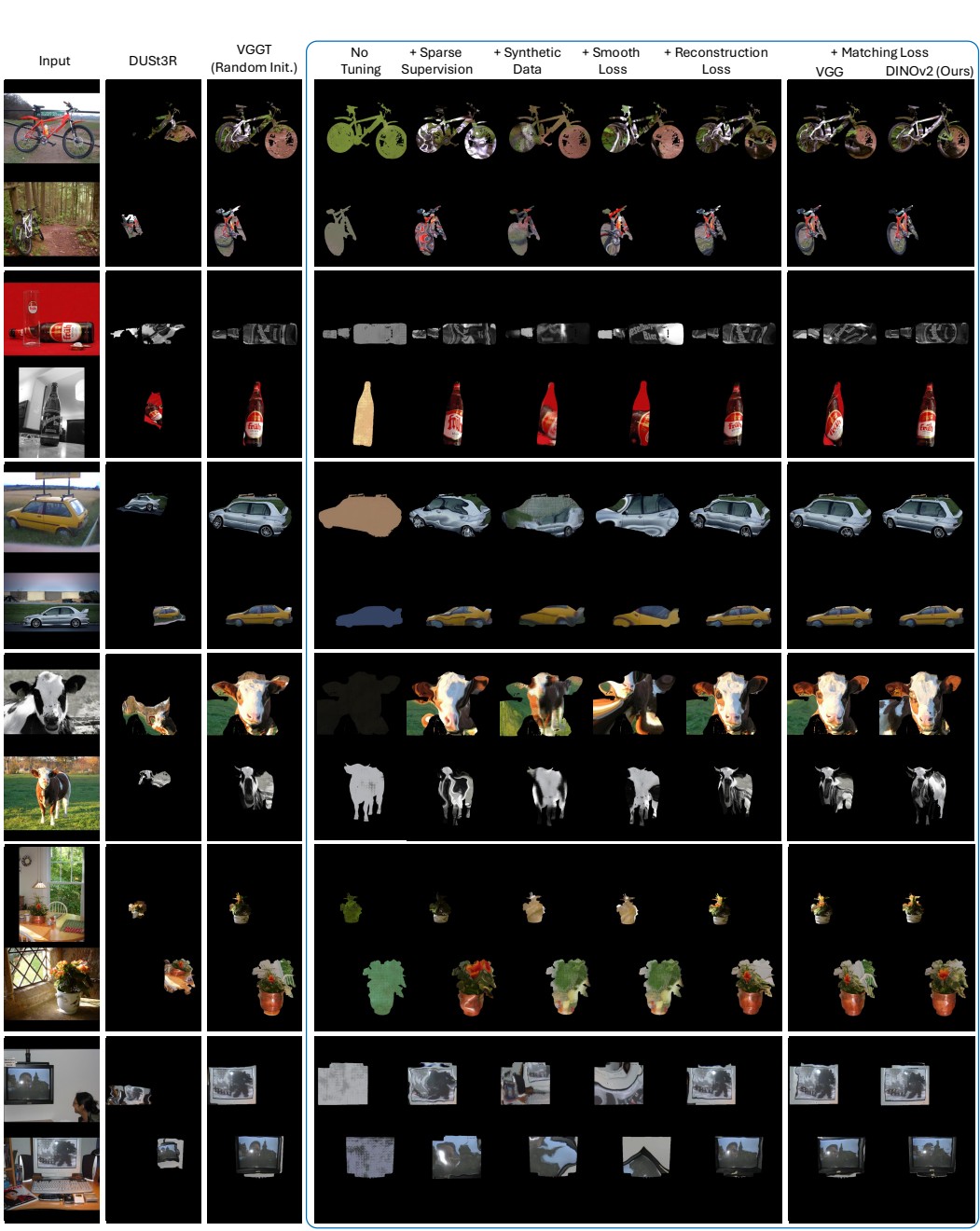

Figure 26: **More qualitative ablation results of the proposed approach (2).**

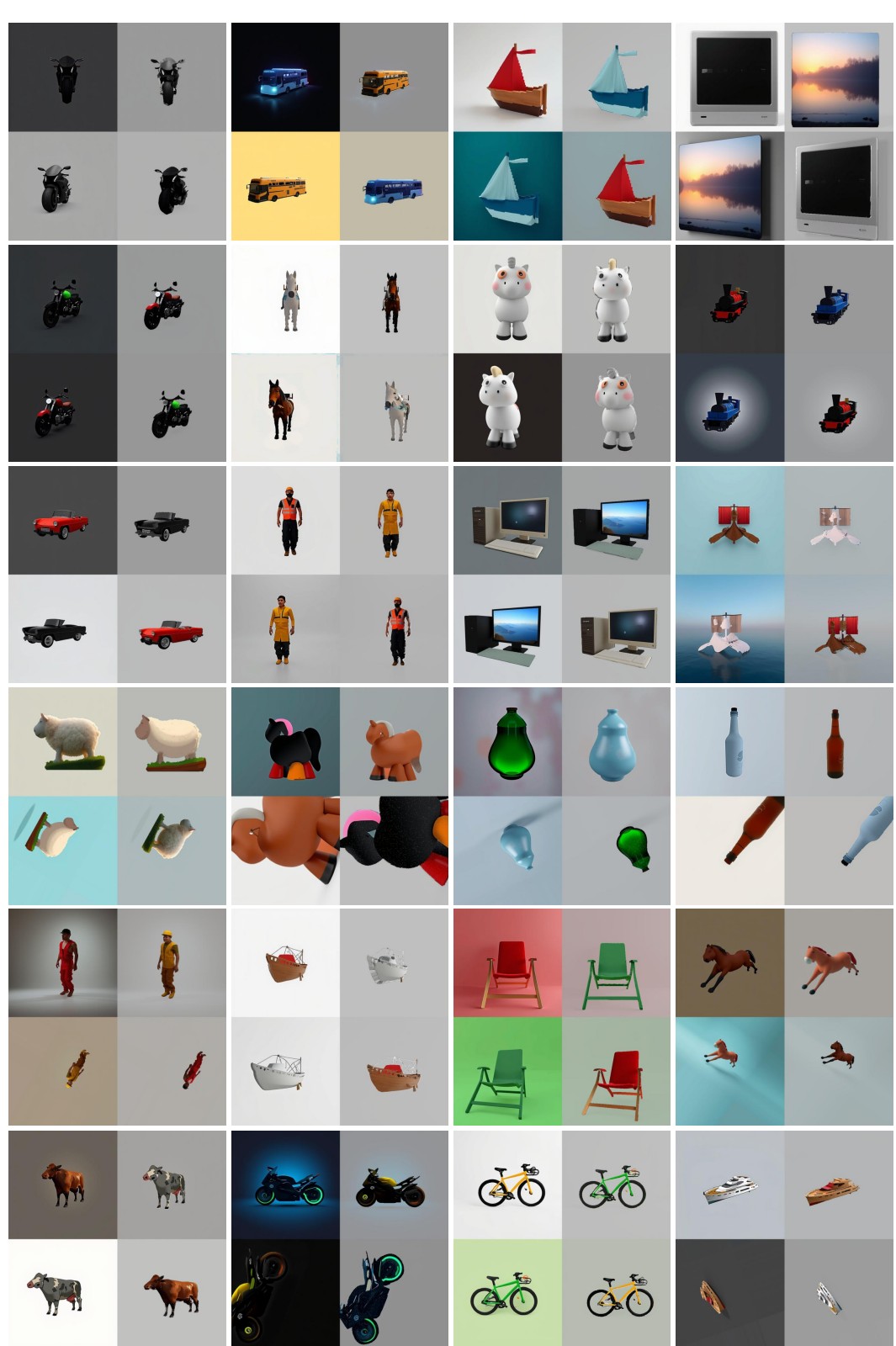

Figure 27: **The paired data sampled from our synthetic dataset.**

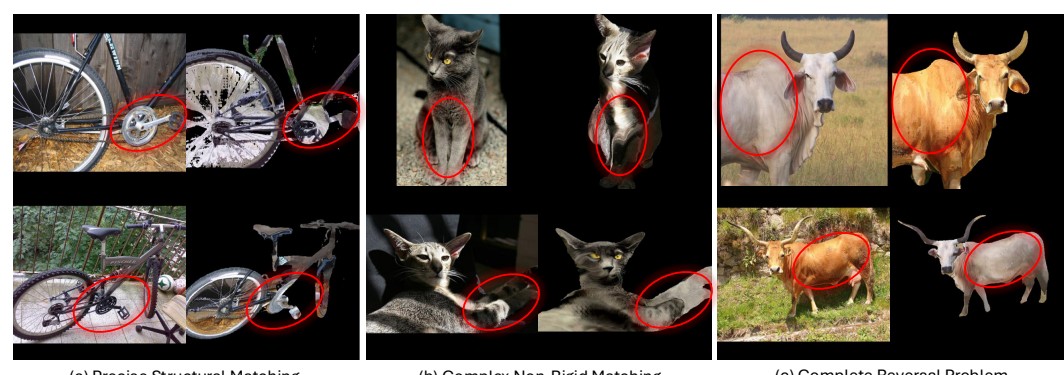

(a) Precise Structural Matching    (b) Complex Non-Rigid Matching    (c) Complete Reversal Problem

Figure 28: **The failure cases of our approach.**

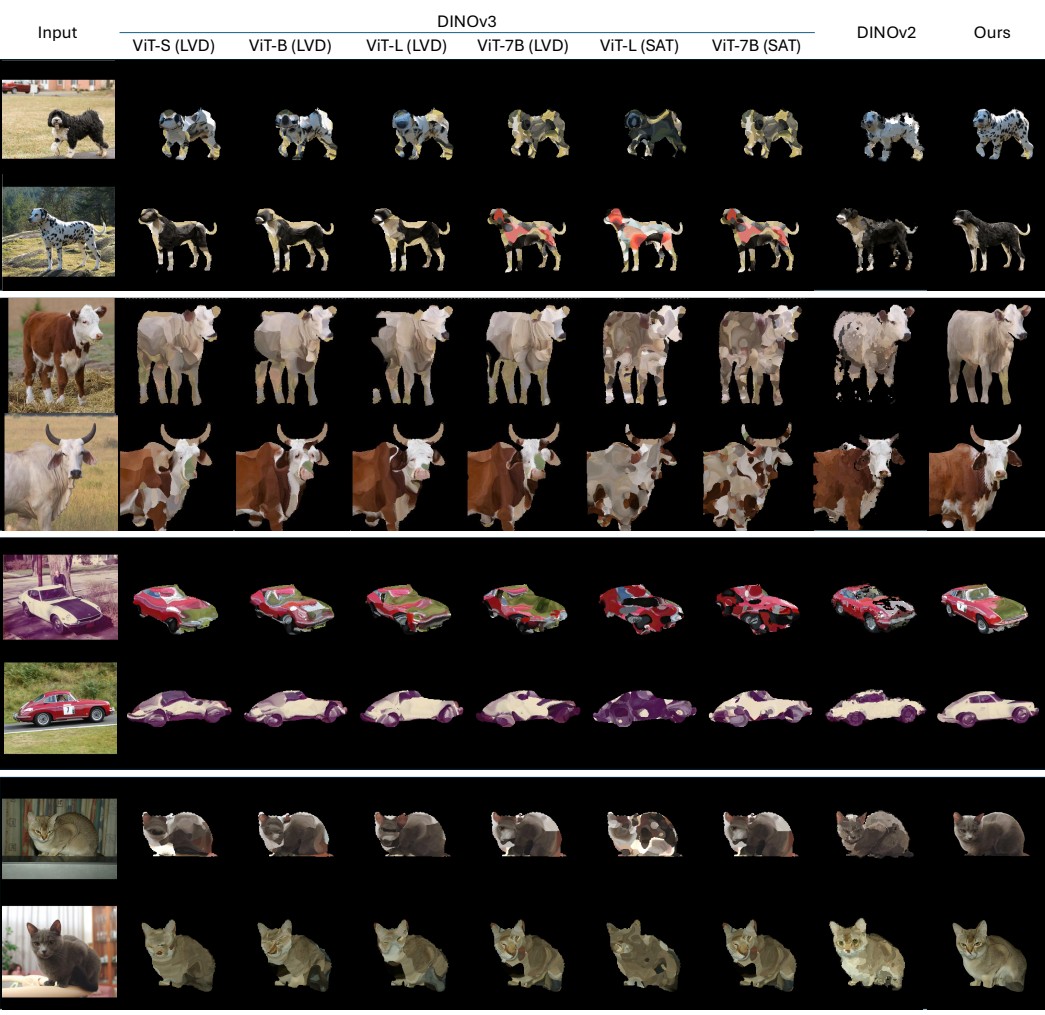

Figure 29: **Qualitative results of using DINOv3 for dense semantic matching.** The ViT-S/B/L/7B represent Small/Base/Large/7B ViT models. The LVD and STA represent web dataset (LVD-1689M) and satellite dataset (SAT-493M).

