# OpenReview forum: "Dense Semantic Matching with VGGT Prior"
_ICLR.cc/2026/Conference — ICLR 2026 Conference Withdrawn Submission_

### Official Review · Reviewer_USQJ · 2025-10-27

**Soundness:** 3
**Presentation:** 3
**Contribution:** 2
**Rating:** 4
**Confidence:** 4

**Summary:**

This paper transfers the priors of Vision Geometry Generation Transformer (VGGT) to the task of cross-instance dense semantic matching. The method fine-tunes the later semantic layers while retaining the early geometry-aware layers, and adds a bidirectional grid + pixel confidence matching head. During training, it combines cycle consistency (matching ↔ reconstruction) with an error-confidence correlation mechanism, using a small amount of sparse annotations along with synthetic dense supervision. Experiments on SPair-71k, AP-10k, and synthetic data achieve state-of-the-art results, demonstrating.

**Strengths:**

Strengths
1. The paper is well-written and clearly structured, making it easy to follow the methodology and understand the contributions.

2. The proposed method effectively leverages the geometric priors from VGGT, demonstrating improved performance in dense semantic matching tasks.

3. Matching-Reconstruction Consistency is a reasonable and effective training strategy, especially in scenarios with limited dense annotations.

4. Verification on multiple datasets (SPair-71k, AP-10k, synthetic data) shows the method's robustness and generalizability.

**Weaknesses:**

1. Insufficient experimental details, such as the resolution, backbone, and pretraining information of different methods in Table 1 are not provided, making it difficult to determine whether the improvements come from the method itself.

2. The results in Table 1 are inconsistent with those reported in other papers. For example, Geo-SC on SPair-71k should have achieved an mIoU of over 80. As reported in this work [1], many works have already achieved an mIoU of 80+ on Spair-71k.

[1] Semantic Correspondence: Unified Benchmarking and a Strong Baseline.

3. The novelty and necessity are limited, and it is unclear whether Vgg-T can outperform the current strongest pretraining models. For example, DinoV3 can also achieve high performance on SPair-71k with simple fine-tuning. The necessity of using VGGT as the backbone needs to be further justified.

**Questions:**

1. In scenarios with long-tail distributions and extreme deformations, does VGGT have advantages over other backbones?

---

### Official Review · Reviewer_Mu5k · 2025-10-28

**Soundness:** 3
**Presentation:** 3
**Contribution:** 1
**Rating:** 4
**Confidence:** 4

**Summary:**

This paper addresses the challenge of dense semantic matching by adapting the 3D geometric foundation model, VGGT. The proposed method preserves geometric features by freezing the early blocks of VGGT, fine-tunes the later blocks, and introduces a semantic head to predict per-pixel correspondences. Training is conducted using a cycle-consistent scheme, synthetic datasets, and a four-stage progressive curriculum. Evaluated on SPair-71k and AP-10K benchmarks, the approach outperforms other methods. The authors claim improvements in geometric disambiguation, manifold preservation, and overall reliability.

**Strengths:**

(1) This paper employs the popular VGGT model for semantic matching, exploring the feasibility of transferring geometric priors to cross-instance, pixel-level correspondences. This effort provides interesting insights into semantic correspondence.
(2) The manuscript is well-structured. Figures and notation are precise, enabling smooth comprehension.

**Weaknesses:**

The paper lacks a compelling motivation for applying VGGT, particularly in justifying the need for a 3D-aware visual backbone in the context of 2D matching. The technical novelty remains limited, as most of the reported improvements derive from standard backbone fine-tuning and cycle-consistency training. Moreover, the work underutilizes VGGT’s broader potential, such as its unique 3D geometric priors and generalization capabilities. The research could be expanded to 2D-to-3D correspondence, rather than being restricted to 2D matching through mere VGGT-based technique transfer.

**Questions:**

1. Could the authors clarify why employing a 3D-aware model such as VGGT is necessary for a 2D semantic matching task? What specific limitations of conventional 2D features does it address or overcome?
2. The VGGT-generated point clouds presented in Figure 1 appear blurry and lack fine-grained details, which seems inconsistent with the precision required for pixel-level correspondence.
3. This work applies VGGT to semantic alignment; however, the authors should elaborate on the substantive novelty or distinctive advantages of using VGGT beyond straightforward technique transfer.
4. How does the strategy of reusing early features, fine-tuning later layers, and incorporating a semantic head differ from conventional CNN or Transformer fine-tuning practices?
5. VGGT is known for its zero-shot generalization capability. Does fine-tuning the backbone compromise this property? Please include performance results under zero-shot semantic matching conditions.
6. It would be beneficial to provide a runtime or computational complexity analysis for semantic matching. How does the inference efficiency compare with that of existing baselines?

---

### Official Review · Reviewer_kes6 · 2025-10-29

**Soundness:** 3
**Presentation:** 3
**Contribution:** 3
**Rating:** 4
**Confidence:** 4

**Summary:**

This paper presents an approch to the classical problem of dense semantic matching leveraging VGGT features and using a separate semantic matching head for actually establishing correspondences, plus a cycle consistency training strategy to improve robustness.

**Strengths:**

The proposed approach is reasonable if not particularly novel, with a reasonable architecture including VGGT features, but also a matching head and a training procedure including cycle consistency (the latter perhaps being the most original part of the approach) and it also appears to give good results in quantitative comparative experiments. Extra training samples are generated using text-to-3D model technology, which may improve the results.

**Weaknesses:**

Although the proposed approach is reasonable, its presentation perhaps overly emphasizes the use of VGGT features: There is no obvious principled reason why a method designed to predict the 3D structure of a single scene for multiple images, and thus presumably encoding some implicit correspondences between these images, should be appropriate for predicting correspondences (or scene 3D structure) from images of different instances of the same category.  This may be reasonable for cars, since they all roughly have the same shape but not obviously so for object categories with widely varying shapes (e.g., mammals) and or configurations (e.g., cats or even people). This issue should be discussed.

**Questions:**

I must confess I did not understand the aliasing part, since the approach is not designed to synthesize pictures but to find correspondences. I didn't understand either what the authors mean by maintain underlying manifold structures during matching. What manifold structures? Finally, it was also unclear to me whether the other methods used as baselines were trained on the same data, which seems a key requirement for fair comparison. The authors should clarify these three points as well as the relevance of VGGT features mentioned in the weaknesses section.

---

### Official Review · Reviewer_YoRu · 2025-11-02

**Soundness:** 2
**Presentation:** 2
**Contribution:** 2
**Rating:** 2
**Confidence:** 4

**Summary:**

This paper adapts VGGT, a 3D geometric foundation model, for dense semantic matching across different instances of the same category. The authors identify two limitations in existing methods: (1) geometric ambiguity in distinguishing symmetric structures, and (2) reliance on nearest-neighbor matching that ignores cross-image invisibility and manifold preservation. To address these, they propose fine-tuning VGGT by freezing early layers, duplicating and fine-tuning later layers, and adding a DPT-based semantic matching head. They introduce a cycle-consistent training strategy with matching-reconstruction consistency and error-confidence correlation, alongside a synthetic data pipeline combining text-to-3D generation with multi-condition image synthesis. Experiments on SPair-71k and AP-10k show 5.2%p improvement over previous SOTA (76.8% vs 71.6% PCK@0.1).

**Strengths:**

1. **Strong empirical results**: Achieves 76.8% PCK@0.1 on SPair-71k (+5.2%p over DIY-SC) with consistent gains across AP-10k benchmarks (Table 1). Comprehensive ablations (Table 3, Fig 6) show VGGT pretraining provides +7.6%p over random initialization.
2. **Manifold preservation**: Qualitative results (Fig 3-4) demonstrate superior preservation of surface geometry compared to baselines, important for downstream tasks. Feature visualization (Fig 7) shows semantic branch produces coherent correspondences.
3. **Confidence prediction**: Error-confidence correlation (Eq 7-9) addresses cross-image invisibility, providing reliability estimates for downstream applications (Fig 5).

**Weaknesses:**

1. **Orthogonal contributions presented as unified framework**: The paper's title emphasizes "VGGT Prior" but the work actually contains two independent contributions: (a) VGGT for geometric ambiguity (Sec 3.1), and (b) cycle-consistent training for annotation scarcity (Sec 3.2). Section 3.2 explicitly states cycle consistency addresses "scarcity of large-scale annotations" and "cross-image invisibility"—neither relates to geometric ambiguity. The two could function independently (cycle consistency with DINO, VGGT without cycle consistency), yet the paper presents them as synergistic without demonstrating that synergy. This framing obscures the actual contributions and makes attribution unclear.
2. **Geometric ambiguity claims lack quantitative validation**: The core motivation—"disambiguate symmetric structures" (Abstract, Introduction)—has no quantitative metrics. Only qualitative checkmarks in Fig 3. Required: (a) left/right confusion rate on symmetric objects (e.g., left_eye↔right_eye), (b) symmetric-object subset evaluation, (c) comparison with 2D methods on symmetric-specific test cases. Table 4 actually shows *high* performance on symmetric objects (Chair 79.1%, Motorbike 77.2%) but *low* on complex articulated ones (Potted plant 30.7%, Person 55.6%), contradicting the geometric ambiguity framing.
3. **Insufficient benchmarks for 3D geometric awareness validation**: Only 2 object-centric datasets (SPair-71k, AP-10k) are used. To validate that improvements stem from "3D geometric foundation model" capabilities, need: (a) diverse categories beyond 18 SPair-71k classes, (b) explicit symmetric structure test set, (c) cross-domain generalization tests, (d) human pose benchmarks (despite "Person" in SPair-71k), (e) 3D-specific evaluation metrics (geodesic distance preservation, surface smoothness). Current evaluation cannot disentangle whether gains come from 3D priors or simply better features.
4. **Unfair baseline comparison prevents attribution**: Current: [VGGT + proposed training] vs [DINO + DIY-SC training]. Missing: (a) VGGT + DIY-SC training, (b) DINO + proposed training. Without these, cannot determine if 5.2%p gain comes from backbone or training methodology. MASt3R comparison also needed as closest 3D geometric foundation model baseline.
5. **Synthetic data reproducibility crisis**: Number of pairs, generation cost, Trellis/FLUX parameters, quality control process—all unreported. Appendix A.7 shows examples but no statistics. This prevents reproduction and cost-benefit assessment.
6. **Statistical significance absent**: No standard deviations, confidence intervals, or multiple runs for the claimed 5.2%p improvement.

### Minor Issues:

1. **Component contributions unclear**: Table 3 shows synthetic data *decreases* performance (75.3%→75.0%), smoothness loss decreases further (→74.2%). Progressive training necessity claimed based on "instability" but no joint training experiment provided.
2. **Computational cost unreported**: No inference time, memory footprint, or FLOPs comparison with baselines—critical for practical utility assessment.

**Questions:**

**Critical :**

1. **MASt3R/DUSt3R comparison**: What is their performance on SPair-71k? While not designed for semantic matching, they represent the closest 3D geometric foundation model baselines. How does your approach differ beyond task formulation?
2. **Fair ablation for attribution**: What is the performance of (a) VGGT + DIY-SC training, and (b) DINO + proposed training? Essential to isolate backbone vs training contributions.
3. **Geometric ambiguity quantification**: Can you provide: (a) left/right confusion rates on symmetric objects, (b) performance on symmetric-object subset vs asymmetric subset, (c) geometric-specific metrics beyond PCK?
4. **Orthogonal contributions**: Can you demonstrate that VGGT and cycle consistency provide synergistic benefits? What is the performance of (a) VGGT without cycle consistency, (b) cycle consistency with DINO backbone?
5. **Synthetic data details**: How many pairs generated? Total GPU cost? Can you release generation code or detailed parameters?

**Secondary:**

1. Why does "No Tuning" VGGT achieve only 9.0% (Table 3) if VGGT has inherent cross-instance alignment (Fig 8)? This suggests fine-tuning, not VGGT priors, is the primary contribution.

---

### Note · Authors · 2025-11-12

I have read and agree with the venue's withdrawal policy on behalf of myself and my co-authors.